# CoCoEmo: Composable and Controllable Human-Like Emotional TTS via Activation Steering

**Siyi Wang** [1]  **Shihong Tan** [2]  **Siyi Liu** [3]  **Hong Jia** [4]  **Gongping Huang** [2]  **James Bailey** [5]  **Ting Dang** [1]

## Abstract

Emotional expression in human speech is nuanced and compositional, often involving multiple, sometimes conflicting, affective cues that may diverge from linguistic content. In contrast, most expressive text-to-speech (TTS) systems enforce a single utterance-level emotion, collapsing affective diversity and suppressing mixed or text–emotion–misaligned expression. While activation steering via latent direction vectors offers a promising solution, it remains unclear whether emotion representations are linearly steerable in TTS, where steering should be applied within hybrid TTS architectures, and how such complex emotion behaviors should be evaluated. This paper presents the first systematic analysis of activation steering for emotional control in hybrid TTS models, introducing a quantitative, controllable steering framework, and multi-rater evaluation protocols that enable composable mixed-emotion synthesis and reliable text–emotion mismatch synthesis. Our results demonstrate, for the first time, that emotional prosody and expressive variability are primarily synthesized by the TTS language module instead of the flow-matching module, and also provide a lightweight steering approach for generating natural, human-like emotional speech.

## 1. Introduction

Human speech conveys affective meaning that extends beyond linguistic content. In natural communication, emotional expression arises from the interaction of internal states, pragmatic intent, and verbal expression, which are often only partially aligned (Keltner et al., 2019). For instance, a speaker may say "I'm fine" while their voice shows frustration. Such example illustrates mixed emotions where multiple affective tendencies coexist, and misaligned emotion where vocal cues diverge from textual meaning. Such phenomena is not incidental: psychological studies also indicate that ambiguity, tension, and partial disclosure are central to perceived naturalness and in human communication (Cabanac, 2002). These findings suggest that emotional expression in natural speech is inherently complex, often combining multiple concurrent and potentially conflicting affective signals rather than a single coherent state.

Despite advances in expressive TTS, most systems assume that emotion can be represented as a single, globally coherent state applied uniformly across an utterance (Im et al., 2022; Tang et al., 2024). For example, a TTS model conditioned on "happy" will typically produce speech that is cheerful from start to finish, even if the text or context naturally suggests mixed feelings, such as "I'm glad you came, but I wish it weren't so late", where a human speaker might express warmth and mild frustration simultaneously. Similarly, subtle mismatches between text and affect, such as nervously laughing while delivering disappointing news, are flattened in most models. This limitation arises from the strong conditional inductive bias imposed by static, utterance-level emotional conditioning (Chandra et al., 2024). Under this bias, the conditional distribution over latent acoustic trajectories is narrowed: reducing expressive capacity in the latent space and causing a functional collapse of affective variability. As a result, mixed emotions are averaged into a single dominant tone, and text–emotion misalignment is suppressed.

Increasing label granularity or retraining models with richer emotion annotations does not resolve this fundamental limitation (Bott et al., 2024; Cho et al., 2024), as long as emotion is encoded as a global and utterance-level condition that enforces emotional coherence by design. In contrast, expressive emotional speech arises from modulating *how* linguistic content is realized, not from conditioning on a single target state. This requires the mechanisms that can reshape acoustic trajectories (e.g., prosody, timing) without redefining the conditional distribution of the model, allow-

---

[1]The University of Melbourne, Australia [2]Wuhan University, China [3]The Hong Kong University of Science and Technology (Guangzhou), China [4]The University of Auckland, New Zealand [5]Monash University, Australia. Correspondence to: Siyi Wang <siyi.wang.4@student.unimelb.edu.au>.

*Proceedings of the $43^{rd}$ International Conference on Machine Learning*, Seoul, South Korea. PMLR 306, 2026. Copyright 2026 by the author(s).

ing multiple, potentially competing affective tendencies to coexist and enabling controlled divergence between textual semantics and acoustic expression.

Steering vectors provide a principled mechanism to achieve this form of latent modulation. Rather than introducing additional conditioning variables or retraining the model, they operate directly in the latent representation space of a pretrained TTS system (Turner et al., 2024; Rimsky et al., 2024; Xie et al., 2025), applying controlled directional biases. Crucially, steering vectors do not require the model to resolve emotional ambiguity into a single representation. Instead, mixed emotional expressions emerge naturally from the quantitative combination of multiple emotion-specific steering directions, while text–emotion misalignment can be expressed by modulating acoustic features independently of textual content. It provides a lightweight and general mechanism for generating emotionally nuanced speech that closely mirrors human expressive behavior.

Despite the promise of steering vectors for controlling emotional expression in TTS models, several fundamental questions remain unresolved.

**Where to steer?** Modern TTS architectures, such as modular text-to-speech language model (SLM) followed by flow-based decoder, learn complex latent representations. It remains unclear which subspaces or layers are most amenable to effective emotion modulation.

**How to steer?** The mechanism of steering has not been systematically explored: how to interpolate directional biases to produce mixed emotions while preserving linguistic content for text-emotion misalignment.

**How to evaluate?** There is no established framework for quantifying the effectiveness of steering vectors for mixed emotions and misaligned text–emotion scenarios. Standard metrics, such as single-label emotion classification, fail to capture multi-dimensional affective coexistence or controlled misalignment.

To address these gaps, this paper makes three key contributions. First, we provide the *first in-depth analysis of emotion representations across modular emotional TTS architectures*, examining both the SLM and flow-matching module. We systematically identify which module, and within it, which layers and operator types (e.g., residual streams, attention outputs), exhibit the highest emotion separability and steering potential, revealing the latent structure of these modular systems. Second, we introduce a method for extracting steering vectors that enables *quantitative mixed-emotion synthesis* through compositional combination and text-independent acoustic steering. This approach allows affective expression to diverge from textual semantics without retraining, supporting flexible and fine-grained emotional control. Third, we develop *a novel multi-rater evaluation*

*framework* that leverages multi-rater annotations to quantify mixed-emotion synthesis.

Our in-depth analysis reveals that SLM layers encode emotional prosody and expressiveness more distinctly than flow-matching modules. We identify specific mid-to-late layers and operations such as attention outputs that are most effective for steering emotional expression. Experiments across multiple datasets and backbones show that injecting steering vectors at these optimal SLM layers enables reliable mixed-emotion synthesis with quantitative control over emotion proportions, while also improving synthesis under text–emotion mismatch. These findings provide a principled framework for analyzing, controlling, and evaluating emotionally nuanced speech in hybrid TTS systems, bridging the gap between human expressive behavior and the global coherence bias of current models. Code is available at https://github.com/wsssy/CoCoEmo.

## 2. Analyzing Emotion in SLM and Flow-Matching

This section addresses the "where to steer" question by analyzing latent representations of emotion. We first compare how emotional information is encoded in the SLM and flow-matching modules, then perform a layer- and operation-level analysis to find the sites with higher linear emotion separability, for more effective steering and emotional control.

**Model Overview.** As illustrated in Figure 1 (left), modern hybrid TTS systems adopt a two-stage architecture. Let $\mathbf{x}_i$ denote the $i^{th}$ input text sequence and $\mathbf{c}_{ref}$ denote the conditioning reference signal for target emotion, which may include reference audio or explicit emotion descriptors. In the first stage, a text-to-speech language model $f_{\text{SLM}}$ maps these inputs to a sequence of discrete speech tokens $\mathbf{z}_i$,

$$\mathbf{z}_i = f_{\text{SLM}}(\mathbf{x}_i, \mathbf{c}_{ref}), \tag{1}$$

where $\mathbf{z}_i = (z_i^1, \ldots, z_i^T)$ represents the token sequence.

In the second stage, the flow-matching acoustic model $f_{\text{Flow}}$ transforms the speech token sequence into a mel-spectrogram, which is then converted to a waveform by a pretrained vocoder $g_{\text{voc}}$:

$$\mathbf{m}_i = f_{\text{Flow}}(\mathbf{z}_i, \mathbf{c}_{ref}), \quad \mathbf{y}_i = g_{\text{voc}}(\mathbf{m}_i). \tag{2}$$

### 2.1. Where to Steer I: Modular Analysis

This modular two-stage design raises a central question: *does emotional expression predominantly originate from the discrete token-level representation $\mathbf{z}_i$ produced by $f_{\text{SLM}}$, or from the continuous acoustic transformation implemented by the flow-matching decoder $f_{\text{Flow}}$?*

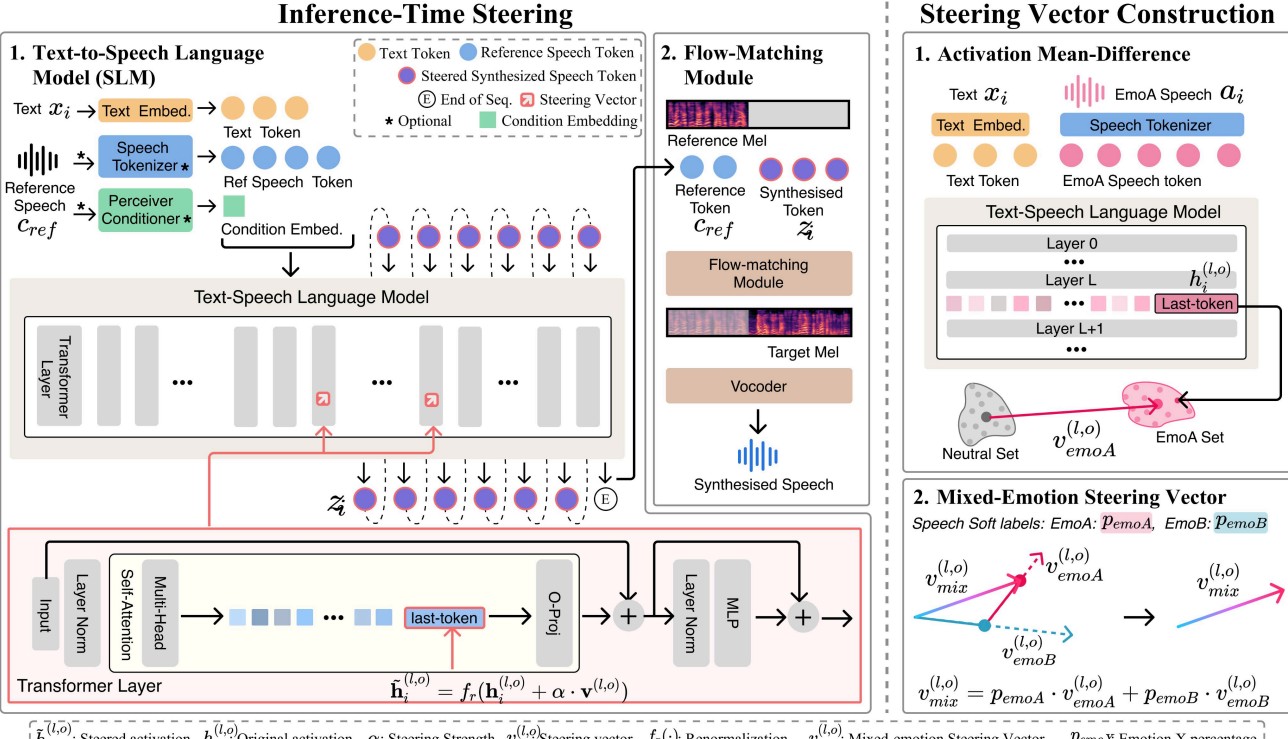

*Figure 1.* **Overview of our method. Left:** Stage-1: The SLM generates speech tokens; steering vectors are injected at selected layers and operators. Stage-2: Flow-matching and vocoder produce the final waveform. **Right:** Steering vector construction.

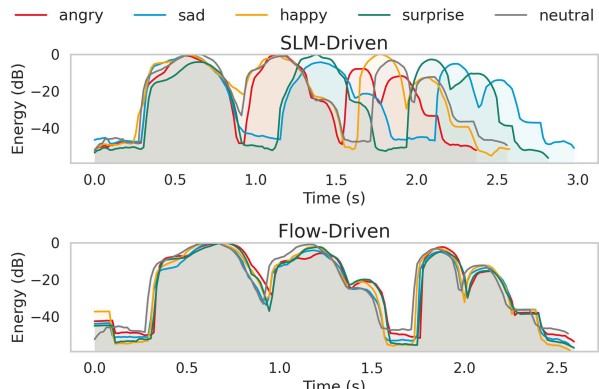

*Figure 2.* Energy contours under cross-conditioning. **Top (SLM-driven)** produce distinct prosodic patterns across emotions; **Bottom (Flow-driven)** yield largely overlapping contours.

**Cross-Conditioning Diagnostic.** To analyze how the Text-to-Speech LM (SLM) and flow-matching (Flow) module each contribute to emotional expression, we design a cross-conditioning diagnostic. Let $\mathbf{c}^e$ and $\mathbf{c}^n$ denote emotional and neutral conditioning signals, respectively.

- **SLM-driven:** Emotion reference is applied only at the SLM to modify speech tokens $\mathbf{z}_i$, while the flow-matching module operates under a neutral condition.

$$\mathbf{z}_i^e = f_{\text{SLM}}(\mathbf{x}_i, \mathbf{c}^e), \quad \mathbf{m}_{\text{SLM}} = f_{\text{Flow}}(\mathbf{z}_i^e, \mathbf{c}^n) \quad (3)$$

- **Flow-driven:** The SLM is neutral, and emotion reference is introduced only via flow-matching.

$$\mathbf{z}_i^n = f_{\text{SLM}}(\mathbf{x}_i, \mathbf{c}^n), \quad \mathbf{m}_{\text{Flow}} = f_{\text{Flow}}(\mathbf{z}_i^n, \mathbf{c}^e) \quad (4)$$

If emotion is mainly encoded in the SLM, SLM-driven conditioning should produce stronger emotional expressiveness; otherwise, flow-driven conditioning should dominate.

For evaluation, we synthesize waveforms from $\mathbf{m}_{\text{SLM}}$ and $\mathbf{m}_{\text{Flow}}$ using the vocoder and measure three emotion-correlated acoustic dimensions: fundamental frequency ($F_0$), energy, and speaking rate (SR). Analyses are conducted separately for *anger*, *sadness*, *happiness*, *neutral*, and *surprise*. We measure prosody consistency using concordance correlation coefficient (CCC) for F0 and energy contours, and quantify the standard deviation (STD) of SR across emotions. Lower CCC and higher STD indicate greater acoustic differentiation, reflecting stronger emotional variability in the synthesized speech (Appendix A.2).

**Findings and Design Implications.** Figure 2 visualizes energy contours of synthesized speech across the five emotions. In the SLM-driven condition, contours diverge markedly, producing distinct prosodic patterns across emotions. In the Flow-driven condition, contours largely overlap, suggesting that the flow-matching module does not alter the prosody but mainly performs acoustic rendering.

*Table 1.* Cross-conditioning diagnostic on CosyVoice2 ($N$=300). Values are mean $\pm$ std.

| CONDITION | $F_0$ CCC ↓ | ENERGY CCC ↓ | SR STD ↑ |
|---|---|---|---|
| SLM-DRIVEN | 0.109 | 0.308 | 0.691 |
| FLOW-DRIVEN | 0.305 | 0.737 | 0.518 |

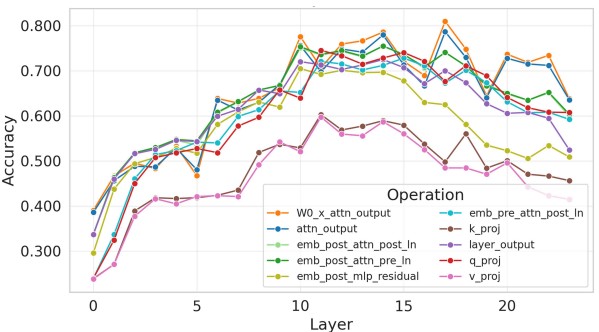

*Figure 3.* Emotion discriminability across layers and operations (CosyVoice2). Mid-to-late layers and attention show high separability.

Table 1 further presents quantitative observations. The Flow-driven condition shows higher CCC for $F_0$ and energy and lower SR STD, indicating stronger alignment across emotions, while the SLM-driven condition exhibits lower CCC and higher SR STD, reflecting greater acoustic differentiation. Overall, SLM primarily governs variability in synthesized emotional features, whereas the flow-matching mainly refines local rendering.

---

**Takeaway 1: SLM vs Flow Matching**

SLM is the primary driver of emotional prosody. Thus, emotion steering should be applied at the SLM.

---

### 2.2. Where to Steer II: Layer and Operator Selection

We now address the finer-grained question: *at which layer and operator within the SLM should steering vectors be injected?* Instead of heuristic selection, we adopt a **discriminability-driven** approach to identify layers and operators where emotions are more linearly separable, enabling more reliable extraction of steering vectors.

**Why Linear Separability?** The effectiveness of steering vectors requires classes to be geometrically organized so a single direction reliably shifts activations without affecting other attributes (Park et al., 2024; Torop et al., 2025). For mixed emotions, vectors are expected to be combined to point toward a different direction producing complex expressions. Thus, linear separability naturally proxies steerability: higher separability enables reliable extraction and thus combination of steering vectors for modulation of mixed emotions.

**Layer- and Operation-Level Probing for SLM Steering.** Let $\mathcal{D} = \{(\mathbf{x}_i, \mathbf{a}_i, y_i)\}_{i=1}^N$ be a dataset of $N$ samples,

where $\mathbf{x}_i$ is input text, $\mathbf{a}_i$ the reference emotional speech, and $y_i \in \{0, 1, \ldots, E\}$ the emotion label. The SLM has $L$ transformer layers, each with multiple operations $\mathcal{O}^{(l)}$ (Figure 1). Layer-wise and operational-wise activations are:

$$\mathbf{h}_i^{(l,o)} = \begin{cases} \mathrm{Op}^{(l,o)}(\mathbf{x}_i, \mathbf{a}_i), & l = 1, \\ \mathrm{Op}^{(l,o)}(\mathbf{h}_i^{(l-1)}), & l = 2, \ldots, L, \end{cases} \quad o \in \mathcal{O}^{(l)}, \quad (5)$$

where $\mathrm{Op}^{(l,o)}$ denotes operations such as attention and feed-forward networks (Appendix A.3). We use last-token representation that summarizes cumulative emotional encoding.

To identify where emotion is most distinctly represented, we train linear probes on $\mathbf{h}_i^{(l,o)}$ to predict $y_i$, measuring linear separability via accuracy. Top-$K$ layers and operations with highest discriminability are used to extract and inject steering vectors, enabling precise layer- and operation-level emotional control.

**Findings and Design Implications.** As shown in Figure 3, layers 10–17 generally exhibit the strongest linear separability in CosyVoice2. Among operations, attention outputs (`attn_output`) consistently achieve the highest discriminability. While the precise peak varies by model (layers 10–17 for CosyVoice2 (Du et al., 2024b), layers 5–10 for IndexTTS2 (Zhou et al., 2025) (Appendix B for IndexTTS2), mid-to-late layers and attention outputs consistently show the strongest performance.

---

**Takeaway 2: Layer & Operator Selection**

Mid-to-late layers and attention outputs exhibit the highest linear separability of emotion representations.

---

## 3. Proposed Emotion Steering Framework

Based on Section 2, we propose a steering framework for complex emotion synthesis. First, we extract a steering vector for each individual emotion at identified model layers. Mixed-emotion vectors are then formed as weighted combinations of these single-emotion vectors, enabling quantitative control over emotion proportions. Steering vectors are extracted primarily from emotional acoustic variations, independently of linguistic representations, to handle text–emotion mismatches.

### 3.1. Steering Vector Construction

**Single-Emotion Steering.** As shown in Figure 1 right, we compute emotion steering vectors using a mean-difference approach, from the mean neutral representation to the mean target emotion representation. To isolate acoustic emotion information, we compare only samples with the same speaker and transcript, ensuring that the resulting vector captures primarily the acoustic emotional variations in the latent space, rather than differences due to content or speaker.

Given the dataset $\mathcal{D} = \{(\mathbf{x}_i, \mathbf{a}_i, y_i)\}_{i=1}^N$, where $y_i \in$

$\{0, \ldots, E\}$ denotes the emotion label and $y_i = 0$ corresponds to neutral speech, we control for speaker and linguistic content by constructing speaker-matched neutral–emotion pairs. For each target emotion $e \in \mathcal{Y}$, we form two subsets $D^{(e)}$ and $D_0^{(e)}$ by pairing emotion-$e$ utterances with neutral utterances from the same speaker, matching transcripts when available, ensuring that representation differences primarily reflect acoustic and emotional variation.

Let $\mathbf{h}_i^{(l,o)}$ be the last-token activation of sample $i$ at the selected $l$ layer and operation $o$ (Figure 1 right). The steering vector for emotion $e$ is then defined as the difference between the mean representations of emotion-$e$ samples and their paired neutral counterparts:

$$\mathbf{v}_e^{(l,o)} = \frac{1}{|\mathcal{D}^{(e)}|} \sum_{i \in \mathcal{D}^{(e)}} \mathbf{h}_i^{(l,o)} - \frac{1}{|\mathcal{D}_0^{(e)}|} \sum_{j \in \mathcal{D}_0^{(e)}} \mathbf{h}_j^{(l,o)}. \quad (6)$$

The vector $\mathbf{v}_e^{(l,o)}$ captures a direction in the latent space associated with emotion $e$ and can be injected during inference to induce the target emotional expression (see Appendix D.1 for dataset-specific details). Analogous findings in text LLMs suggest that emotional representations tend to be directionally encoded and steerable in latent space (Reichman et al., 2025), supporting the validity of this contrastive extraction approach. In mismatch scenarios where the reference audio emotion differs from the text, the steering vector acts as an internal bias to override the text-implied emotion (see Appendix E.4).

**Mixed-Emotion Steering.** For mixed emotions, we compute a steering vector by combining the single-emotion vectors $\mathbf{v}_e^{(l,o)}$, as shown in Figure 1 bottom right. Let the corresponding weights for target emotions be $\{p_e\}_{e=1}^{E}$, with $\sum_{e=1}^{E} p_e = 1$. The mixed-emotion steering vector is:

$$\mathbf{v}_{\text{mix}}^{(l,o)} = \sum_{e=1}^{E} p_e \mathbf{v}_e^{(l,o)}. \quad (7)$$

### 3.2. Inference-Time Steering

During inference, either the single-emotion steering vector $\mathbf{v}_e^{(l,o)}$ or the mixed-emotion vector $\mathbf{v}_{\text{mix}}^{(l,o)}$ is injected into the selected top-$K$ layers and operations. At each selected layer or operation, the activation $\mathbf{h}$ is modulated via steering:

$$\tilde{\mathbf{h}}_i^{(l,o)} = \mathbf{h}_i^{(l,o)} + \alpha \cdot \mathbf{v}^{(l,o)} \quad (8)$$

where $\alpha$ controls the steering intensity, and $\mathbf{v}^{(l,o)}$ is either $\mathbf{v}_e^{(l,o)}$ (single emotion) or $\mathbf{v}_{\text{mix}}^{(l,o)}$ (mixed emotions). To preserve the original activation scale and maintain semantic coherence, we renormalize as $\tilde{\mathbf{h}}_i^{(l,o)} \leftarrow \frac{\|\mathbf{h}_i^{(l,o)}\|}{\|\tilde{\mathbf{h}}_i^{(l,o)}\|} \cdot \tilde{\mathbf{h}}_i^{(l,o)}$.

### 3.3. Mixed-Emotion Evaluation

Evaluating mixed-emotion synthesis requires a soft ground truth. We leverage multi-rater annotations: each speech recording $\mathbf{a}_i$ is labeled by $M$ raters with one-hot vectors $y_{i,m} \in \{0,1\}^{|E|}$. The consensus distribution is

$$\boldsymbol{p}_i = \frac{1}{M} \sum_{m=1}^{M} y_{i,m}, \quad (9)$$

(e.g., for $E = \{\text{happy}, \text{sad}, \text{angry}\}$, two raters label *happy* and one labels *sad*, yielding $\boldsymbol{p}_i = [\frac{2}{3}, \frac{1}{3}, 0]$). These consensus distributions serve as mixing weights for steering vector $\mathbf{v}_{\text{mix}}^{(l,o)}$ in Eq. (7). Synthesized speech is then compared with the groundtruth target speech $\mathbf{a}_i$ where $\boldsymbol{p}_i$ is derived, providing a robust evaluation of mixed-emotion.

## 4. Experiments

### 4.1. Experimental Setup

**Datasets.** For steering-vector extraction, we combine ESD (Zhou et al., 2021), RAVDESS (Livingstone & Russo, 2018), and CREMA-D (Cao et al., 2014), which provide matched linguistic content in different emotional styles, encouraging vectors that capture acoustic variations while preserving content. The combined dataset contains 20,691 utterances (around 4,000 per emotion) with a speaker-independent train/validation/test split of 0.5/0.2/0.3.

Evaluation covers in-distribution (ID) and out-of-distribution (OOD) settings. Mixed-emotion synthesis uses 772 CREMA-D ID samples and 1,055 IEMOCAP OOD samples, while text–emotion mismatch uses 2,164 IEMOCAP samples. Details are in Appendix D.1.

**Backbones and Baselines.** We evaluate our activation steering on two representative state-of-the-art hybrid TTS systems: CosyVoice2 (Du et al., 2024b) and IndexTTS2 (Zhou et al., 2025), which represent the strongest i) instruction-based and ii) explicit vector-based TTS systems (see Appendix A.1). For all tasks, we use a neutral reference speech to ensure that synthesized emotions reflect only the effect of the steering vectors.

We compare against several strong and diagnostic baselines:

- **No-steering**: Original model outputs.
- **Random-noise steering**: Inject a random vector with the same magnitude as the steering vector at the same sites.
- **Dominant-emotion steering**: Inject only the dominant emotion vector scaled to the match magnitude.
- **Instruction-based control** (CosyVoice2): Use natural-language prompts for emotion control. Instruction1 specifies qualitative blends; Instruction2 specifies quantitative percentages based on soft labels (Appendix E.1).
- **Emotion-vector control** (IndexTTS2): use the built-in emotion-weight vector for control, with scaling set to 0.6, the maximum that preserves speech intelligibility and speaker characteristics (Appendix E.1; see Appendix E.3 for a full scaling sweep).

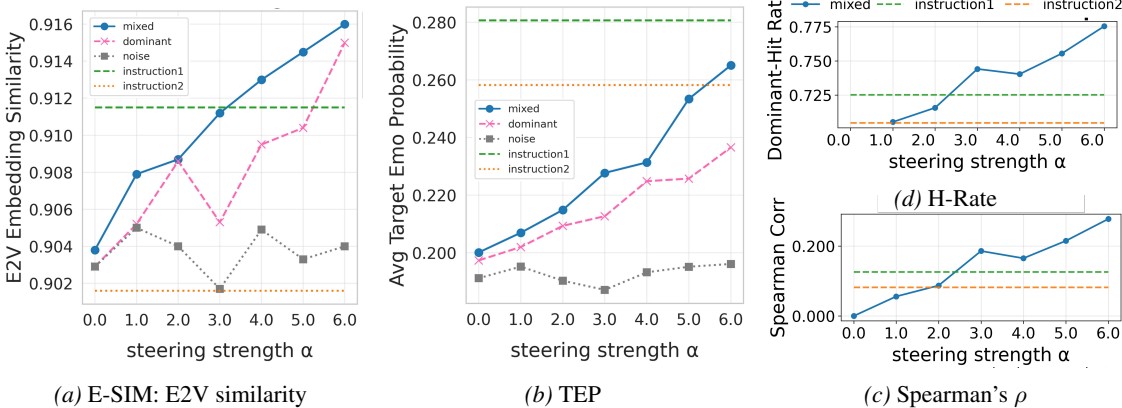

*Figure 4.* Mixed-emotion synthesis results for CosyVoice2 on IEMOCAP. Higher values indicate better performance.

**Text–Emotion Mismatch Severity.** We measure text and audio emotions independently and quantify their mismatch using the valence–arousal (VA) distance on IEMOCAP. Text VA values are predicted using a text emotion classifier (Ma et al., 2024), and audio VA values are taken from dataset annotations. We compute the $\ell_2$ distance $d$ between text and audio VA coordinates and partition samples into three mismatch levels: **low** ($d < 0.14$), **mid** ($0.14 \leq d < 0.42$), and **high** ($d \geq 0.42$) (see Appendix E.4).

**Evaluation Metrics.** *Objective evaluation* include:

- *Emotion Controllability:* **E-SIM** (Emotion Similarity), cosine similarity between Emotion2Vec (E2V) embeddings (Ma et al., 2024) of synthesized and target speech; **TEP** (Target Emotion Probability), probability assigned to the target emotion by an emotion classifier (Ma et al., 2024); for mixed emotions, **Spearman Correlation** ($\rho$) (Spearman, 1961) measures rank agreement between predicted and ground-truth emotion order, and **H-Rate** (Dominant-Hit Rate) is the fraction of samples where the dominant emotion shows the largest relative probability increase. Higher values indicate better control.
- *Speaker Consistency:* **S-SIM**, cosine similarity between WavLM-base (Chen et al., 2022) speaker embeddings of synthesized and reference speech.
- *Speech Intelligibility:* **WER** (Word Error Rate), measured using Whisper-Large-V3 (Radford et al., 2023).

*For subjective evaluation*, we conduct Mean Opinion Score (MOS) tests, with **Naturalness MOS (N-MOS)** assesses perceived speech quality by 10 to 13 humans.

### 4.2. Mixed-Emotion Speech Synthesis

Figure 4(a)–(b) shows mixed-emotion synthesis results. Compared to random-noise steering, both mixed- and dominant-emotion steering consistently guide speech toward target emotions, confirming the effectiveness of the steering vectors. E-SIM and TEP are higher for mixed-

than dominant-emotion steering, indicating successful control over the compositional mixed directions. Increasing $\alpha$ further improves alignment with target emotions.

Interestingly, emotion control baselines using Instruction1 outperforms Instruction2, suggesting that quantitative prompt control struggles more than qualitative control within CosyVoice2 itself. Steering offers finer-grained flexibility across $\alpha$ values, yielding higher E-SIM and TEP at larger $\alpha$, though TEP does not surpass Instruction1. Further analysis with H-Rate and Spearman correlation $\rho$ in Figure 4(c) shows that steering aligns synthesized emotions with the ground-truth mixed-emotion ranking quantitatively, whereas instruction-based control, despite higher TEP, biases toward mixed directions without proportional quantitative control. Similar trends are observed across other backbones and datasets (Appendix F).

Table 2 reports quantitative results for mixed-emotion control on CREMA-D (ID) and IEMOCAP (OOD) across multiple backbones. Applying steering vectors directly to the backbone (CoCoEmo with varying $\alpha$) consistently improves E-SIM, H-Rate, and correlation $\rho$ over no-steering and emotion-control baselines (instruction-based and emotion-vector), demonstrating effective and quantitative mixed-emotion control. Speaker similarity (S-SIM) is preserved, and WER remains comparable with only minor degradation. N-MOS results further show that perceived speech quality is maintained or improved on in-distribution data. On OOD data, the slight N-MOS reduction reflects a well-documented expressivity–naturalness trade-off, explicitly calibrated via $\alpha$.

We further demonstrate that activation steering is a flexible plug-and-play enhancement for existing emotion-control methods (CoCoEmo with Ins/Emo_V with $\alpha$). Adding steering on top of instruction-based or emo_vector conditioning yields additional gains in TEP, $\rho$, and H-Rate without degrading quality in general. While E-SIM increases on CREMA-D over steering alone, we observe a slight de-

*Table 2.* Mixed-emotion evaluation on CREMA-D. Objective metrics (E-SIM, TEP, $\rho$, H-Rate, S-SIM, WER) and subjective metric (N-MOS) are reported. Best results are **bold**, second-best are underlined.

| Backbone | Model | E-SIM↑ | TEP↑ | $\rho$↑ | H-Rate↑ | S-SIM↑ | WER↓ | N-MOS↑ |
|---|---|---|---|---|---|---|---|---|
| **In-distribution evaluation on CREMA-D** | | | | | | | | |
| | No-steer | 0.743 | 0.065 | - | - | 0.871 | 1.07 | 4.11 |
| | Instruction1 (Ins1) | 0.761 | 0.200 | 0.111 | 0.694 | 0.853 | 0.22 | 4.11 |
| | Instruction2 (Ins2) | 0.762 | 0.169 | 0.104 | 0.688 | 0.851 | **0.06** | 3.36 |
| CosyVoice2 | CoCoEmo ($\alpha$=3.0) | 0.762 | 0.100 | 0.166 | 0.709 | **0.872** | 1.01 | **4.25** |
| | CoCoEmo ($\alpha$=5.0) | 0.779 | 0.149 | 0.209 | 0.724 | 0.870 | 0.78 | 3.96 |
| | CoCoEmo (Ins1, $\alpha$=3.0) | 0.780 | 0.291 | 0.225 | 0.728 | 0.849 | 0.27 | 3.38 |
| | CoCoEmo (Ins1, $\alpha$=5.0) | 0.790 | **0.335** | **0.319** | **0.760** | 0.846 | 0.20 | 3.00 |
| | CoCoEmo (Ins2, $\alpha$=3.0) | 0.777 | 0.266 | 0.207 | 0.726 | 0.852 | 0.53 | 3.42 |
| | CoCoEmo (Ins2, $\alpha$=5.0) | **0.795** | 0.315 | 0.297 | 0.755 | 0.845 | 0.91 | 3.66 |
| | No-steer | 0.751 | 0.037 | - | - | **0.864** | 5.72 | 3.82 |
| | Emo-Vector (Emo_V) | 0.767 | 0.165 | 0.236 | 0.731 | 0.850 | 5.74 | 4.05 |
| IndexTTS2 | CoCoEmo ($\alpha$=3.0) | 0.767 | 0.090 | 0.072 | 0.681 | **0.864** | 5.83 | 3.95 |
| | CoCoEmo ($\alpha$=5.0) | **0.782** | 0.140 | 0.240 | 0.734 | 0.863 | 5.81 | 4.13 |
| | CoCoEmo (Emo_V, $\alpha$=3.0) | 0.770 | 0.247 | 0.324 | 0.760 | 0.845 | 5.85 | **4.66** |
| | CoCoEmo (Emo_V, $\alpha$=5.0) | 0.774 | **0.278** | **0.399** | **0.789** | 0.836 | 5.89 | 4.16 |
| **Out-of-Distribution (OOD) evaluation on IEMOCAP** | | | | | | | | |
| | No-steer | 0.903 | 0.197 | - | - | 0.888 | 6.70 | 3.94 |
| | Instruction1 (Ins1) | 0.911 | 0.280 | 0.126 | 0.725 | 0.874 | 2.41 | 3.88 |
| | Instruction2 (Ins2) | 0.902 | 0.258 | 0.082 | 0.705 | 0.876 | 2.85 | **4.06** |
| CosyVoice2 | CoCoEmo ($\alpha$=3.0) | 0.911 | 0.228 | 0.186 | 0.744 | **0.891** | 5.86 | 3.72 |
| | CoCoEmo ($\alpha$=5.0) | **0.915** | 0.253 | 0.215 | 0.755 | 0.890 | 6.27 | 3.91 |
| | CoCoEmo (Ins1, $\alpha$=3.0) | 0.906 | 0.316 | 0.188 | 0.746 | 0.872 | **2.38** | 2.18 |
| | CoCoEmo (Ins1, $\alpha$=5.0) | 0.898 | **0.349** | **0.297** | **0.783** | 0.870 | 2.70 | 2.54 |
| | CoCoEmo (Ins2, $\alpha$=3.0) | 0.900 | 0.313 | 0.201 | 0.752 | 0.876 | 2.51 | 3.40 |
| | CoCoEmo (Ins2, $\alpha$=5.0) | 0.894 | 0.343 | 0.294 | 0.778 | 0.873 | 3.78 | 3.15 |
| | No-steer | 0.885 | 0.187 | - | - | 0.876 | 5.65 | **4.37** |
| | Emo-Vector (Emo_V) | 0.855 | 0.296 | 0.254 | 0.767 | 0.854 | **4.74** | 4.15 |
| IndexTTS2 | CoCoEmo ($\alpha$=3.0) | **0.892** | 0.227 | 0.201 | 0.752 | **0.883** | 5.06 | 4.33 |
| | CoCoEmo ($\alpha$=5.0) | 0.884 | 0.251 | 0.284 | 0.778 | 0.882 | 5.20 | 4.32 |
| | CoCoEmo (Emo_V, $\alpha$=3.0) | 0.844 | 0.311 | **0.326** | **0.791** | 0.843 | 5.25 | 4.12 |
| | CoCoEmo (Emo_V, $\alpha$=5.0) | 0.824 | **0.315** | 0.282 | 0.775 | 0.833 | 6.15 | 4.06 |

*Table 3.* Evaluation on the high-mismatch set of IEMOCAP. Best results are **bold**, second-best are underlined.

| Method | E-SIM↑ | TEP↑ | S-SIM↑ | WER↓ | N-MOS↑ |
|---|---|---|---|---|---|
| **CosyVoice2** | | | | | |
| No-steer | 0.802 | 0.197 | **0.896** | 9.18 | 4.22 |
| Instruction | 0.843 | 0.436 | 0.887 | **4.01** | 4.23 |
| CoCoEmo ($\alpha$=3.0) | 0.833 | 0.332 | 0.895 | 8.03 | 4.29 |
| CoCoEmo ($\alpha$=6.0) | **0.862** | **0.504** | 0.892 | 8.74 | **4.40** |
| **IndexTTS2** | | | | | |
| No-steer | 0.825 | 0.318 | 0.885 | **5.13** | 4.17 |
| Emo_vector | 0.872 | 0.667 | 0.877 | 5.75 | 4.05 |
| CoCoEmo ($\alpha$=3.0) | 0.857 | 0.478 | 0.884 | 5.34 | 4.08 |
| CoCoEmo ($\alpha$=6.0) | **0.874** | **0.681** | **0.886** | 6.58 | 4.21 |

crease on IEMOCAP, which is likely due to the subtler emotional expressions in the IEMOCAP dataset, where overly emotion control reduces the embedding similarity slightly. Overall, activation steering provides an effective and flexible mechanism for mixed-emotion control in hybrid TTS, regardless of whether original emotion conditioning is present. We also compare with EmoSteer-TTS (Xie et al., 2025), the only comparable training-free method, which steers the flow-matching module; CoCoEmo achieves comparable and stronger mixed-emotion control while better preserving speaker similarity (Appendix F.1). We further verify that simultaneous steering in both SLM and flow-matching modules degrades performance, confirming the advantage of targeted SLM steering (Appendix F.2).

### 4.3. Text–Emotion Mismatch Speech Synthesis

Figure 5 shows E-SIM under low, mid, and high text–emotion mismatch for CosyVoice2. Baseline "Instruction" conditions follow target emotion instructions mismatched with the text. Performance declines as mismatch increases, indicating text guidance alone cannot override intrinsic text-implied biases. Activation steering consistently improves E-SIM, with larger $\alpha$ yielding greater gains, showing effective biasing toward the desired acoustic emotion.

This trend holds across backbones (Table 3). Even for In-

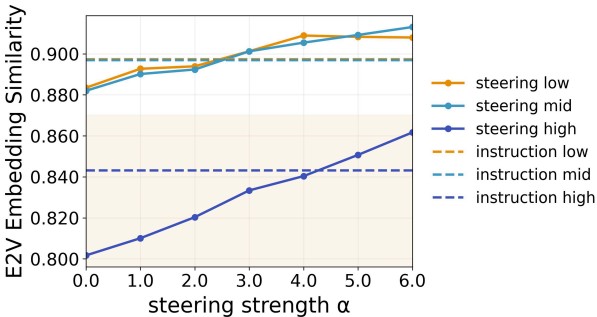

*Figure 5.* E-SIM of high text–emotion mismatch synthesis under low, mid, and high conditions using CosyVoice2 on IEMOCAP.

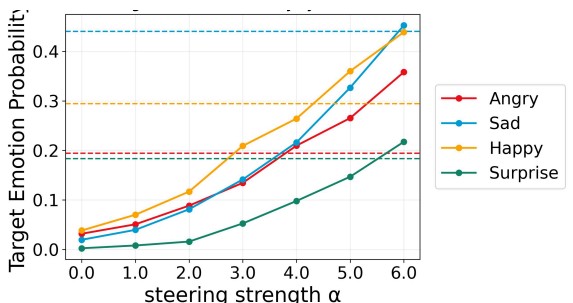

*Figure 6.* TEP for single-emotion synthesis on CosyVoice2 using ESD, RAVDESS, and CREMA-D. Dashed lines: instruction baselines.

dexTTS2, which is specifically designed for explicit emotional conditioning, steering provides complementary E-SIM and TEP gains, while also preserves speaker similarity (S-SIM) and maintains comparable WER. The improved subjective N-MOS also confirms the perceived speech quality (Appendix G).

### 4.4. Single-Emotion Steering

We further validate the effectiveness of single-emotion steering, which forms the foundation for successful mixed-emotion steering. Here, we apply single-emotion steering vector and evaluate its effect for each target emotion separately. As shown in Figure 6 on ESD, RAVDESS, and CREMA-D, TEP increases with larger $\alpha$ compared to the no-steering ($\alpha = 0$) and instruction baselines, indicating a correct directional bias introduced by the steering vectors. Similar trends are observed on out-of-distribution data and other backbones (see Appendix H).

### 4.5. Steering Control Across Layers

**Layer-wise Steering Analysis** To validate that higher linear separability in latent representations enables more effective steering (Section 2), we perform layer-wise steering analysis. Figure 7 shows TEP peaks at layers 17 and 14 for CosyVoice2, matching the layers with highest separability in Figure 3. Overall, layer-wise TEP correlates with separability (correlation $\rho = 0.5078$), supporting that layers with

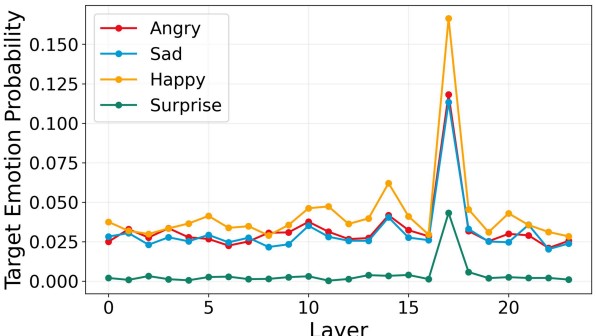

*Figure 7.* Layer-wise TEP across SLM layers in CosyVoice2 using ESD, RAVDESS, and CREMA-D ($\alpha = 3.5$).

higher linear separability enable more effective steering and reliable mixed-emotion superposition, which also holds for IndexTTS2 (Appendix C).

**Hyperparameter Selection.** To determine the optimal number of steering layers $K$, we evaluate configurations using the top-1 to top-5 layers based on discriminative analysis and layer-wise steering effects. Considering the trade-off between emotion controllability and speech intelligibility (see ablation in Appendix I), we finally select the top-2 layers (layers 17 and 14) for CosyVoice2 and the top-3 layers (layers 6, 8, and 1) for IndexTTS2. Empirically, we find that $\alpha \in [0, 4.5]$ provides a stable operating range without degrading quality, while larger values up to $\alpha = 6.0$ may occasionally reduce intelligibility.

## 5. Related Work

**Controllable Emotional Speech Synthesis.** Recent expressive TTS systems predominantly adopt hybrid pipelines that combine language models with flow-matching or diffusion-based decoders (Du et al., 2024a;b; Zhou et al., 2025; Chen et al., 2025; Eskimez et al., 2024; Zhang et al., 2025; Ji et al., 2024), achieving remarkable zero-shot quality but relying on implicit prompt or reference alignment for emotion control.

Several methods introduce fine-grained interfaces, including few-shot intensity adjustment, emotion-adaptive vectors, style diffusion, and preference optimization (Chen et al., 2024; Cho et al., 2025; Li et al., 2023; Gao et al., 2025a), yet synthesizing natural human emotional expressions, especially mixed emotions quantitatively or in mismatch scenarios, remains challenging due to the inherent limitations of global conditional vectors. While prompt-based approaches can provide instructions for mixed emotions (Zhou et al., 2022; Gao et al., 2025b) or enforce target mismatched emotions, they struggle with quantitative control of emotion proportions and fail to overcome dominant text-implied emotion biases (Peng et al., 2025). In contrast, we directly perturb the latent space, steering acoustic directions to

enable composable mixed-emotion control and mismatched emotions synthesize.

**Activation Steering.** Activation steering modulates model behavior by adding direction vectors to intermediate activations at inference time, assuming high-level concepts admit approximately linear representations (Turner et al., 2024; Zou et al., 2023; Rimsky et al., 2024). Recent work has primarily focused on LLMs, where advanced methods explore preference-optimized learned vectors, disentangled injection, and input-dependent prediction (Cao et al., 2024; Torop et al., 2025; Parekh et al., 2025; He et al., 2025), demonstrating effective modulation of various high-level concepts. Despite these successes in text LLMs, it remains unclear whether the linear separability assumption holds in TTS, or whether activation steering can effectively control emotional expression, given the fundamentally different multimodal training, continuous acoustic representations, and the entanglement of speaker, semantic content, and emotion in TTS. In particular, mixed-emotion synthesis and text–emotion mismatch scenarios pose additional challenges: it is unknown whether compositional steering vectors for mixed emotions, or the disentanglement of text and acoustic information in the latent space, can enable reliable synthesize.

## 6. Conclusion

This paper presents a systematic study of activation steering in hybrid TTS models, demonstrating that steering provides an efficient and flexible mechanism to control emotional expression in synthesized speech. We show that steering enables fine-grained mixed-emotion generation, guides emotional expression toward intended acoustic directions independent of linguistic content, effectively handling text–emotion mismatches. These findings pave the way for future TTS systems where model behavior can be perturbed to produce richer, more natural, and human-like emotional speech, with potential applications in mental health, affective computing, and other human-centered technologies.

**Limitations and Future Work.** Several directions remain for future investigation. First, the steering vector is applied uniformly across tokens. While each steered token causally influences subsequent tokens through autoregressive attention (Turner et al., 2024; Park et al., 2024), explicit segment-level steering could enable finer-grained temporal control over dynamic emotional shifts within an utterance. Second, discrete emotion labels are required only during steering vector extraction; at inference, the extracted vectors support continuous intensity interpolation and compositional blending at arbitrary weights. Extending the framework to continuous affective dimensions such as valence–arousal could enable more fine-grained emotional control following the structure of human affect. Third, the current framework does not support free-form text prompts as a control

interface. A lightweight mapping from text descriptions to soft emotion proportions could bridge this gap. Fourth, our evaluation combines objective metrics with subjective naturalness ratings. Future work could further include subjective evaluation of perceived emotion faithfulness to provide a more comprehensive assessment. Finally, the geometry of emotion representations in TTS latent spaces remains underexplored. Future work could probe whether more complex nonlinear structures exist using tools such as sparse autoencoders or manifold analysis, potentially leading to more effective steering operations.

## Impact Statement

This work improves controllability in neural text-to-speech by enabling fine-grained and compositional emotional steering (including mixed-emotion interpolation) without task-specific retraining. Such capability can benefit assistive communication (e.g., more expressive speech for users with speech impairments), education and storytelling, and human–computer interaction where prosody and affect help convey intent and reduce misunderstanding. It may also reduce the need for repeated fine-tuning, potentially lowering engineering and compute overhead.

However, stronger emotional control in synthetic speech can increase misuse risks. In particular, it can make impersonation and social-engineering attacks more persuasive by adding emotionally appropriate prosody to fabricated audio. Voice cloning and voice deepfakes have been highlighted as enabling fraud and scams and are often difficult for humans to reliably detect. We therefore emphasize responsible deployment: systems built on this technique should require consent for any voice identity that resembles a real person, include user-facing disclosure that speech is synthetic, and implement abuse prevention mechanisms (e.g., usage policies, rate limits, monitoring, and, where feasible, technical provenance such as watermarking or metadata). Disclosure guidelines for synthetic voices can help design transparent user experiences. We also note that emotion expression is culturally and demographically variable; biases in emotion-labeled data could lead to uneven performance or stereotyped affect, so future work should audit across populations and contexts.

## Acknowledgements

We thank the anonymous reviewers for their constructive feedback that helped improve this paper. This research was supported by the University of Melbourne's Research Computing Services and the Supercomputing Center of Wuhan University.

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

# A. Implementation Details

## A.1. TTS Backbone Configurations

We evaluate our method on two state-of-the-art LM-based TTS systems with different architectural designs. Table 4 summarizes their configurations.

*Table 4.* Architecture configurations of the TTS backbones used in our experiments.

| Configuration | CosyVoice2 | IndexTTS2 |
|---|---|---|
| LM Architecture | Qwen2-based decoder | GPT2-style decoder |
| Number of Layers ($L$) | 24 | 24 |
| Hidden Dimension ($d$) | 896 | 1024 |
| Attention Heads | 14 | 16 |
| Vocabulary Size | 6561 (audio tokens) | 8192 (audio tokens) |
| Operating Space | Text tokens $\rightarrow$ audio codes | Audio codebook tokens |
| Flow-Matching | DiT-based | DiT-based |
| Vocoder | HiFi-GAN | BigVGANv2 |

**Key Differences.** CosyVoice2 employs a pretrained Qwen2-based decoder that processes text tokens to produce high-level linguistic representations, which are subsequently transformed into audio codebook representations by downstream acoustic modules. In contrast, IndexTTS2 directly operates on audio codebook tokens and does not rely on a pretrained text language model for autoregressive generation. This architectural distinction leads to differences in the structure of internal representations and, consequently, in the optimal steering locations and magnitudes for each model.

**Cross-Conditioning Configuration.** Table 5 details how emotion conditioning is applied in the LM and acoustic/flow modules for our cross-conditioning experiments.

*Table 5.* Emotion conditioning mechanisms in the evaluated TTS backbones.

| Backbone | LM-side emotion-carrying conditioning | Flow-side emotion-carrying conditioning | Neutral instantiation |
|---|---|---|---|
| CosyVoice2 | reference speech token | Reference acoustic prompt (mel features), reference speeech token, speaker embedding | LM-side: neutral reference; Flow-side: neutral reference |

Our cross-conditioning protocol swaps these two sources independently: Condition SLM-driven injects emotion only via the LM-side conditioning, while Condition flow-driven injects emotion only via the acoustic-side conditioning. For each comparison, speaker identity and script are fixed within the same group, and only the designated conditioning source is changed.

## A.2. Prosody Consistency via CCC

For each sample, we evaluate two cross-conditioning settings (SLM-driven and Flow-driven) and construct a set of five utterances consisting of four emotion renderings and a base neutral utterance. We extract frame-level F0 and energy contours from each utterance and temporally align them within the sample (by using a common contour length).

We then compute concordance correlation coefficient (CCC) for every unordered pair of utterances in the set and report the **mean pairwise CCC** as the per-sample prosody-consistency score (computed separately for F0 and for energy):

$$\text{CCC}(x, y) = \frac{2\rho_{xy}\sigma_x\sigma_y}{\sigma_x^2 + \sigma_y^2 + (\mu_x - \mu_y)^2}, \qquad \overline{\text{CCC}} = \frac{1}{|\mathcal{P}|} \sum_{(i,j) \in \mathcal{P}} \text{CCC}(x_i, x_j),$$

where $\mathcal{P}$ denotes all unordered pairs among the five utterances. Dataset-level results are summarized by the mean and standard deviation of per-sample $\overline{\text{CCC}}$.

### A.3. Hook Points and Candidate Sites

For each transformer layer $l \in \{1, \ldots, L\}$, we define a set of hook points $\mathcal{H}$ corresponding to key intermediate representations in the self-attention and feed-forward computation, which serve as candidate steering sites:

- `emb_pre_attn_post_ln`: Input to the self-attention block after input layer normalization.

- `q_proj`: Query projection output in self-attention.

- `k_proj`: Key projection output in self-attention.

- `v_proj`: Value projection output in self-attention.

- `attn_output`: Self-attention output before the output projection (`o_proj`).

- `W0_x_attn_output`: Output of the attention output projection (`o_proj`).

- `emb_post_attn_pre_ln`: Post-attention residual stream before layer normalization.

- `emb_post_attn_post_ln`: Post-attention residual stream after layer normalization.

- `emb_post_mlp_residual`: Output of the MLP block after residual addition.

- `layer_output`: Final output of the transformer layer.

IndexTTS2 adopts a GPT2-style attention implementation in which the query, key, and value projections are computed jointly via a single linear transformation; accordingly, we use a single `qkv_proj` hook corresponding to the combined projection as the candidate steering site.

## B. IndexTTS2 Discriminability Results

Figure 8 shows discriminability profiles for IndexTTS2, which exhibits peak separability at earlier layers (5–10) compared to CosyVoice2 (10–17). This architectural difference motivates per-model calibration of steering sites.

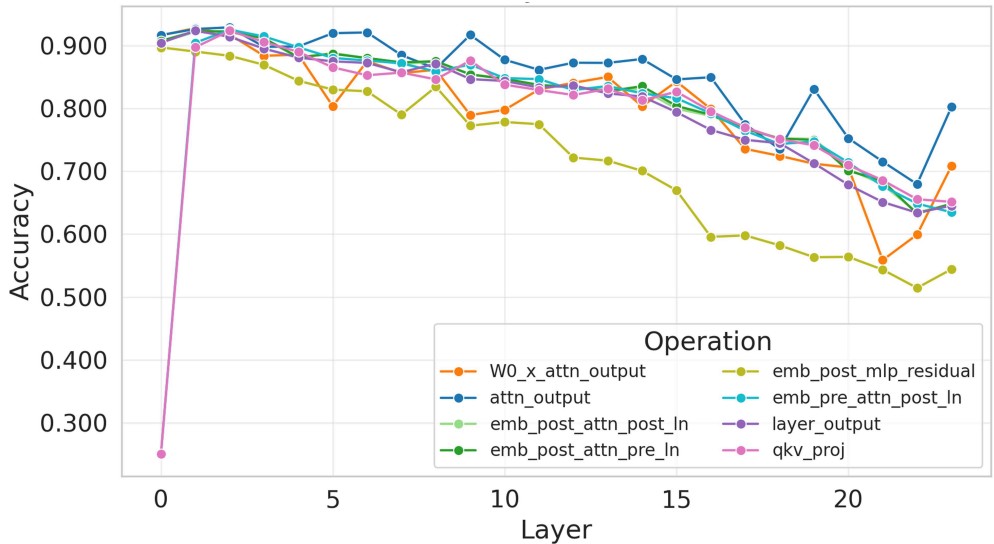

*Figure 8.* IndexTTS2 multi-class discriminability profile. Peak at layers 5–10.

We further analyze steering effectiveness across layers and relate it to linear discriminability results. While shallow layers in IndexTTS2 exhibit high linear classification accuracy for emotion, steering at these layers is relatively ineffective. We attribute this to the presence of explicit emotion conditioning in IndexTTS2, which is injected as an input and strongly influences early representations. Although shallow-layer activations are linearly separable with respect to emotion, adding a

steering vector at these layers constitutes a weak intervention: the steering perturbation is applied only to the last token, while the original emotion condition continues to propagate through subsequent layers and dominates the forward dynamics. As a result, the influence of the steering vector is largely attenuated during generation.

In contrast, mid-layer representations (e.g., layer 6 and 8) not only exhibit strong linear discriminability but also serve as effective intervention points. At these layers, emotional information is more internally consolidated and less directly overridden by the input emotion condition. Consequently, steering vectors applied at mid layers exert a more persistent and causally effective influence on the generated speech. This observation highlights that linear discriminability alone is insufficient to predict steering success; the layer's role in the generative computation and its susceptibility to downstream overwriting are equally critical.

## C. IndexTTS2 Layer-wise Steering Result

We further perform a layer-wise steering analysis to empirically evaluate steering effectiveness across the IndexTTS2 backbone. Figure 9 reports the average target emotion probability (TEP) achieved by applying steering vectors at different layers.

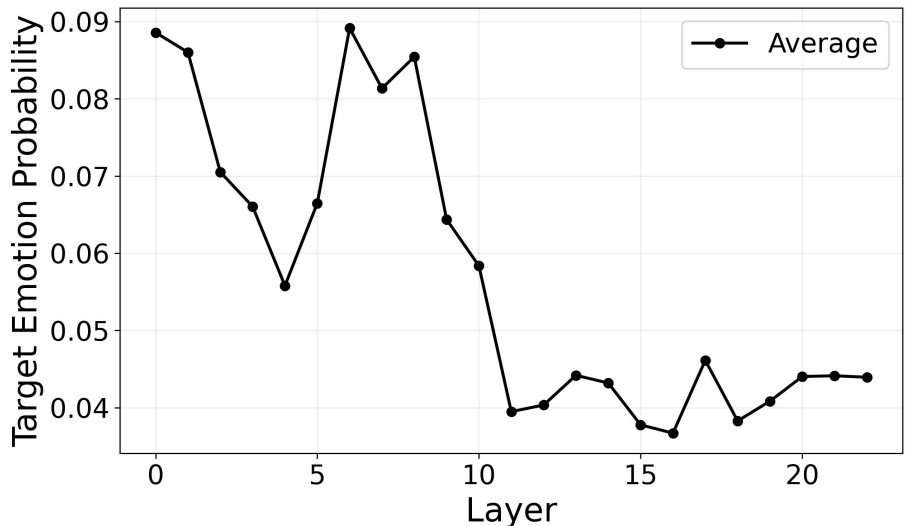

*Figure 9.* Layer-wise average TEP across SLM layers in IndexTTS2 using ESD, RAVDESS, and CREMA-D ($alpha$ = 3.5)

Consistent with the discriminability profile in Figure 8, steering effectiveness peaks at initial and mid layers (notably layers 6 and 8), which exhibit both strong linear separability and high causal influence on the final acoustic realization. However, steering at layer 0 leads to severe degradation in speech intelligibility.

This observation suggests that effective steering layers should not only ensure that (i) emotional information is sufficiently disentangled with high linear separability to enable targeted manipulation, but also that (ii) the layer occurs at a stage of computation where modifications are not immediately overridden by subsequent conditioning mechanisms. Therefore, shallower layers, despite high discriminability, can be dominated by the persistent influence of input-level emotion conditioning. Instead, mid-layer representations in IndexTTS2 satisfy both criteria.

Overall, these results reinforce that while steering performance is governed by linear separability to a high degree, the interaction between representational structure and model-specific generative dynamics also plays a key role.

## D. Datasets

### D.1. Steering Vector Extraction Datasets

We use multiple emotional speech datasets with paired recordings (same speaker, same text, different emotions) for steering vector extraction. Table 6 summarizes the data statistics.

To minimize confounding factors, we use ESD, RAVDESS, and CREMA-D, which are designed to contain parallel utterances

*Table 6.* Statistics of datasets used for steering vector extraction. Numbers indicate the count of utterances per emotion category.

| Dataset | Happy | Neutral | Sad | Surprise | Angry |
|---|---|---|---|---|---|
| ESD | 3500 | 3500 | 3500 | 3500 | 3500 |
| RAVDESS | 192 | 96 | 192 | 192 | 192 |
| CREMA-D | 326 | 1032 | 208 | – | 761 |
| **Total** | 4018 | 4628 | 3900 | 3692 | 4453 |

with the same speaker and transcript expressed under different emotions. This structure allows us to attribute representation differences primarily to emotional variation rather than speaker identity or linguistic content. For CREMA-D, we remove samples whose provided emotion labels are inconsistent with the majority vote of human annotations, and reserve these excluded samples for mixed-emotion synthesis evaluation. This pairing protocol ensures that the difference-in-means operation primarily isolates emotional variation while substantially reducing content variation.

**Data Splits.** We use speaker-disjoint splits with ratios of 50%/20%/30% for train/validation/test. The training set is used for computing steering vector centroids, the validation set for site selection (linear probe evaluation), and the test set for final evaluation. All experiments use English utterances only.

### D.2. Evaluation Datasets

**In-Distribution Evaluation.** For single-emotion steering, we evaluate on the test splits of ESD, RAVDESS, and CREMA-D, which are the same datasets used for steering vector extraction. For mixed-emotion steering, we use CREMA-D samples whose majority-vote human annotations differ from the dataset-provided labels. We further restrict this set to utterances whose annotations lie within {happy, sad, angry, surprised, neutral} and contain more than two non-neutral emotion labels. This yields 772 samples for evaluation.

**Out-of-Distribution (OOD) Evaluation.** To assess generalization, we evaluate on **IEMOCAP**, which consists of conversational dyadic interactions annotated along dimensional affective axes (valence, arousal, dominance). IEMOCAP is used for both OOD evaluation and text–emotion misalignment analysis.

For mixed-emotion experiments, we derive soft emotion distributions from multi-rater annotations by normalizing annotator disagreement. We retain only samples involving the five target emotions, merging *excited* with *happy* and *frustrated* with *angry*. We further filter utterances to durations between 2 and 10 seconds and remove samples containing multiple speakers using pyannote/speaker-diarization-3.1 model, resulting in 1,055 samples.

For text–emotion mismatch evaluation, we select utterances with a single non-neutral emotion label according to multi-rater annotations. These samples are stratified into low-, mid-, and high-mismatch subsets as defined in Appendix E.4. For each emotion–mismatch subset (e.g., *happy–low*), we randomly sample up to 500 utterances. The same dataset is used for single-emotion speech synthesis under mismatch conditions, yielding 2,164 samples in total.

## E. Additional Experimental Details

### E.1. Baseline Models

**Instruction-Based Emotion Control (CosyVoice2).** We evaluate prompt-based emotion control using CosyVoice2's native natural-language instruction interface, without modifying internal model activations. Two instruction variants are considered.

**Instruction1 (Descriptive Instructions).** Emotion is specified using categorical or descriptive prompts. For single-emotion synthesis and text–emotion mismatch experiments, the prompt requests a single target emotion. For mixed-emotion synthesis, the prompt qualitatively describes a blend of multiple emotions without specifying explicit proportions.

**Instruction2 (Percentage-Based Instructions).** Emotion is specified using explicit percentage-based prompts that indicate the relative contribution of multiple emotions. The percentages are derived from the ground-truth emotion distribution obtained from multi-rater annotations. This setting represents an oracle-assisted prompt-based baseline.

**Illustrative Prompt Examples.**    The following examples illustrate the prompt formats used in our experiments:

- *Single-emotion (Instruction1):* "Say it in a happy tone."
- *Mixed-emotion, descriptive (Instruction1):* "Say it in a blend of happy, sad, and angry emotions."
- *Mixed-emotion, percentage-based (Instruction2):* "Say it in 40% happy, 30% sad, and 30% surprise."

**Emotion-vector control (IndexTTS2).**    We evaluate emotion control in IndexTTS2 using its native continuous emotion conditioning interface, without modifying internal model activations. Emotion is specified by directly providing an explicit emo_vector. The emo_vector is an 8-float list specifying the intensity of each emotion in the fixed order *[happy, angry, sad, afraid, disgusted, melancholic, surprised, calm]*.

For single-emotion synthesis, we assign a weight of $0.6$ to the target emotion and zero to all others. For mixed-emotion synthesis, the vector is constructed as $0.6 \cdot \mathbf{p}$, where $\mathbf{p}$ denotes the ground-truth emotion distribution derived from multi-rater annotations. We choose a scaling factor of $0.6$ rather than $1.0$ because stronger conditioning was empirically observed to degrade speech intelligibility in some cases. This value provides a balanced operating point that yields effective emotion control while preserving speech quality.

### E.2. Evaluation Metrics

We use the following metrics for comprehensive evaluation:

**Emotion Similarity (E-SIM).**    Cosine similarity between the Emotion2Vec embedding of the synthesized speech and the ground-truth emotion provided in the dataset. Higher values indicate better emotion matching.

**Target Emotion Probability (TEP).**    Probability assigned to the target emotion by the Emotion2Vec classifier. Higher values indicate stronger perceived target emotion.

**Spearman's Correlation ($\rho$).**    Used to measure rank-based agreement between predicted and ground-truth emotion rankings in mixed-emotion scenarios. Higher values indicate better alignment with human labels.

**Dominant Hit Rate (H-Rate).**    The proportion of samples where the predicted dominant emotion has the highest increased probability. Higher values indicate better alignment with human-annotated dominant labels.

**Speaker Similarity (S-SIM).**    Cosine similarity between WavLM embeddings of synthesized and reference speech. Higher values indicate better speaker consistency. Reported on a scale $[0, 1]$.

**Word Error Rate (WER).**    Computed using Whisper Large-v3 transcriptions compared against the ground-truth text. Lower values indicate better transcription accuracy.

**Mean Opinion Score (MOS).**    Speech *naturalness* is evaluated via human listening tests using a 5-point Absolute Category Rating (ACR) scale ($1 =$ completely unnatural, $5 =$ completely natural). We report the mean MOS across raters (higher is better) (Streijl et al., 2016).

### E.3. Emo-Vector Scaling Sweep for IndexTTS2

To justify the choice of Emo-Vector scaling factor used in the main experiments (Table 2), we report a full sweep over scaling values from 0.1 to 0.8 on both CREMA-D and IEMOCAP for mixed-emotion synthesis.

As shown in Table 7, increasing the scaling factor improves emotion controllability (TEP, $\rho$, H-Rate) but degrades speaker similarity (S-SIM), particularly beyond 0.6. On CREMA-D, S-SIM drops from 0.865 at scale 0.1 to 0.830 at scale 0.8; on IEMOCAP, from 0.882 to 0.833. We select 0.6 as the default, as it provides strong emotion control while maintaining acceptable speaker preservation.

*Table 7.* Emo-Vector scaling sweep on IndexTTS2 for mixed-emotion synthesis.

| Dataset | Scale | E-SIM↑ | TEP↑ | $\rho$↑ | H-Rate↑ | S-SIM↑ | WER↓ |
|---|---|---|---|---|---|---|---|
| CREMA-D | 0.1 | 0.754 | 0.045 | 0.007 | 0.663 | **0.865** | 5.74 |
| | 0.2 | 0.755 | 0.053 | 0.082 | 0.693 | 0.863 | 5.72 |
| | 0.3 | 0.761 | 0.062 | 0.045 | 0.672 | 0.863 | 5.81 |
| | 0.4 | 0.767 | 0.092 | 0.097 | 0.693 | 0.863 | 5.75 |
| | 0.5 | **0.769** | 0.139 | 0.224 | 0.730 | 0.861 | **5.71** |
| | 0.6$^\dagger$ | 0.767 | 0.165 | 0.236 | 0.731 | 0.850 | 5.74 |
| | 0.7 | **0.769** | 0.212 | 0.271 | 0.745 | 0.842 | 5.81 |
| | 0.8 | 0.757 | **0.246** | **0.291** | **0.754** | 0.830 | 5.80 |
| IEMOCAP | 0.1 | 0.891 | 0.202 | 0.085 | 0.715 | **0.882** | 5.24 |
| | 0.2 | **0.893** | 0.219 | 0.155 | 0.738 | 0.881 | 5.07 |
| | 0.3 | 0.889 | 0.245 | 0.184 | 0.746 | 0.877 | 5.06 |
| | 0.4 | 0.884 | 0.266 | 0.260 | 0.768 | 0.870 | 4.94 |
| | 0.5 | 0.869 | 0.286 | 0.265 | 0.770 | 0.862 | **4.51** |
| | 0.6$^\dagger$ | 0.855 | 0.296 | 0.254 | 0.767 | 0.854 | 4.74 |
| | 0.7 | 0.856 | **0.305** | **0.293** | **0.781** | 0.839 | 4.77 |
| | 0.8 | 0.839 | 0.302 | 0.291 | 0.779 | 0.833 | 4.74 |

$^\dagger$ Default scaling used in main experiments.

### E.4. Text-Emotion Mismatch Severity Levels

For text-emotion mismatch experiments on IEMOCAP, we quantify the degree of semantic-acoustic conflict using the $\ell_2$ distance $d$ between predicted text emotion (valence-arousal) and labeled speech emotion. After normalizing both to $[0, 1]$, we partition samples into three severity levels:

- **Low**: $d < 0.1 \times \sqrt{2} \approx 0.14$
- **Mid**: $0.14 \leq d < 0.3 \times \sqrt{2} \approx 0.42$
- **High**: $d \geq 0.3 \times \sqrt{2} \approx 0.42$

Text VA values are predicted using a context-aware valence-arousal regression model (Christ et al., 2024), with outputs normalized to $[0, 1]$. Audio VA values are provided by dataset annotations and mapped from the original $[1, 5]$ scale to $[0, 1]$ via $(v - 1)/4$.

## F. Mixed-Emotion Synthesis Additional Results

This appendix provides supplementary analysis of mixed-emotion steering behavior across different backbones and datasets. We include results for IndexTTS2 on CREMA-D (in-distribution), IndexTTS2 on IEMOCAP (out-of-distribution), and CosyVoice2 on CREMA-D.

In in-distribution settings (CREMA-D), mixed-emotion steering consistently improves E-SIM, TEP, Spearman correlation $\rho$, and Dominant-Hit Rate as the steering strength $\alpha$ increases, and outperforms dominant-only steering for both CosyVoice2 and IndexTTS2 (Figures 11 and 10).

A single exception appears for IndexTTS2 on the out-of-distribution IEMOCAP dataset (Figure 12), where dominant-emotion steering slightly exceeds mixed-emotion steering in E-SIM and larger $\alpha$ does not yield further embedding similarity gains. Nevertheless, TEP and rank-based metrics continue to improve, indicating that emotion control remains effective. This behavior likely reflects distribution shift interacting with IndexTTS2's emotion-conditioning design and does not affect the overall conclusions.

### F.1. Comparison with EmoSteer-TTS

We compare CoCoEmo with EmoSteer-TTS (Xie et al., 2025), the only comparable training-free activation steering method for emotional TTS. EmoSteer-TTS steers the flow-matching module rather than the Text-to-Speech LM. We reproduce EmoSteer-TTS on CosyVoice2 using the same extraction data as CoCoEmo and report results across steering strengths.

As shown in Table 8, CoCoEmo achieves stronger proportional mixed-emotion control than EmoSteer-TTS, with higher $\rho$ (0.209 vs. 0.193 on CREMA-D) and H-Rate (0.724 vs. 0.717; 0.755 vs. 0.755). Crucially, flow-matching steering at

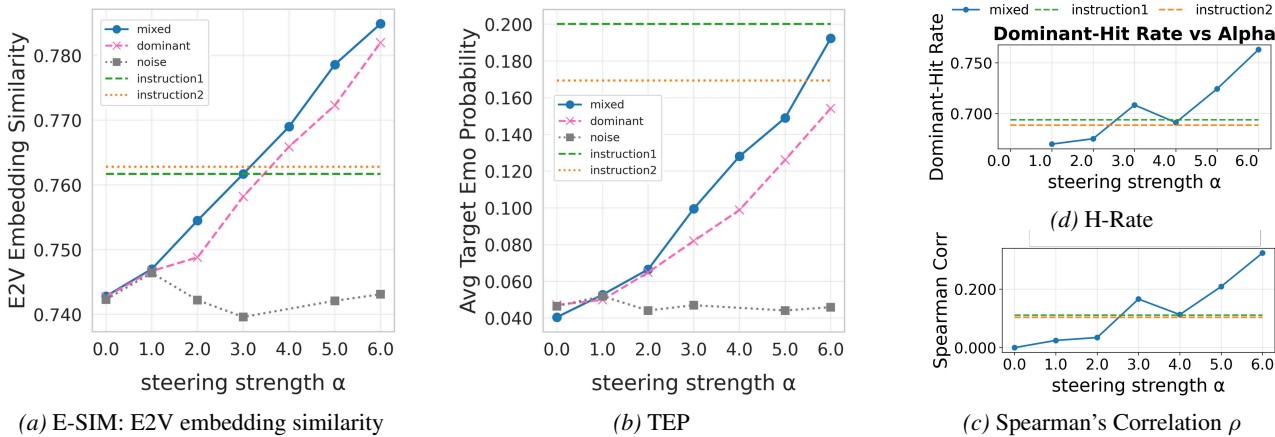

*(a)* E-SIM: E2V embedding similarity      *(b)* TEP      *(c)* Spearman's Correlation $\rho$

*Figure 10.* Mixed-emotion synthesis results for CosyVoice2 using CREMA-D (in distribution) dataset. Higher values indicate better performance.

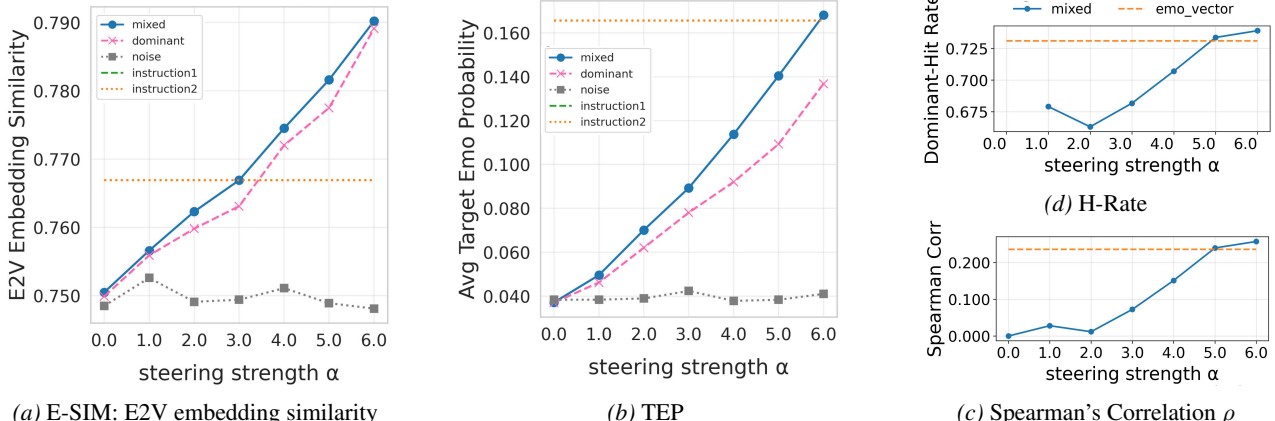

*(a)* E-SIM: E2V embedding similarity      *(b)* TEP      *(c)* Spearman's Correlation $\rho$

*Figure 11.* Mixed-emotion synthesis results for IndexTTS2 using CREMA-D (in distribution) dataset. Higher values indicate better performance.

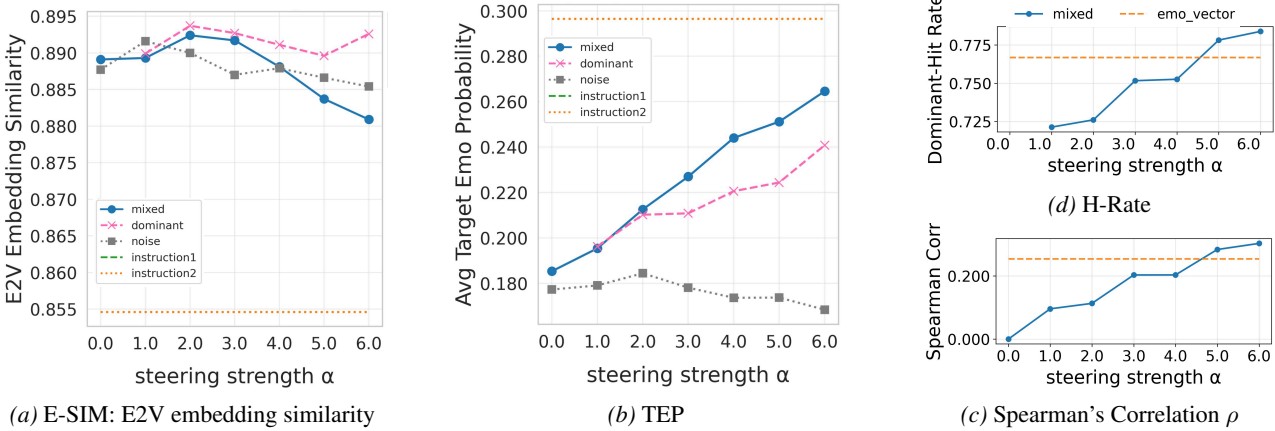

*(a)* E-SIM: E2V embedding similarity      *(b)* TEP      *(c)* Spearman's Correlation $\rho$

*Figure 12.* Mixed-emotion synthesis results for IndexTTS2 using IEMOCAP (out-of-distribution) dataset. Higher values indicate better performance.

*Table 8.* EmoSteer-TTS vs. CoCoEmo on CosyVoice2 for mixed-emotion synthesis.

| Dataset | Method | E-SIM↑ | TEP↑ | $\rho$↑ | H-Rate↑ | S-SIM↑ | WER↓ |
|---|---|---|---|---|---|---|---|
| CREMA-D | No-steer | 0.743 | 0.065 | – | – | 0.871 | 1.07 |
| | EmoSteer $\alpha$=0.5 | 0.749 | 0.058 | -0.013 | 0.659 | 0.867 | 1.10 |
| | EmoSteer $\alpha$=1.0 | 0.767 | 0.097 | 0.098 | 0.691 | 0.858 | **0.76** |
| | EmoSteer $\alpha$=1.5 | 0.780 | **0.167** | 0.072 | 0.676 | 0.836 | 0.83 |
| | EmoSteer $\alpha$=2.0 | **0.786** | 0.160 | 0.193 | 0.717 | 0.807 | 0.79 |
| | CoCoEmo $\alpha$=3.0 | 0.762 | 0.100 | 0.166 | 0.709 | **0.872** | 1.01 |
| | CoCoEmo $\alpha$=5.0 | 0.779 | 0.149 | **0.209** | **0.724** | 0.870 | 0.78 |
| IEMOCAP | No-steer | 0.903 | 0.197 | – | – | 0.888 | 6.70 |
| | EmoSteer $\alpha$=0.5 | 0.903 | 0.199 | -0.004 | 0.686 | 0.890 | 6.33 |
| | EmoSteer $\alpha$=1.0 | 0.910 | 0.218 | 0.138 | 0.729 | 0.885 | 6.08 |
| | EmoSteer $\alpha$=1.5 | 0.913 | 0.257 | **0.216** | **0.755** | 0.869 | 6.33 |
| | EmoSteer $\alpha$=2.0 | 0.909 | **0.272** | 0.117 | 0.721 | 0.844 | 6.15 |
| | CoCoEmo $\alpha$=3.0 | 0.911 | 0.228 | 0.186 | 0.744 | **0.891** | **5.86** |
| | CoCoEmo $\alpha$=5.0 | **0.915** | 0.253 | 0.215 | **0.755** | 0.890 | 6.27 |

higher $\alpha$ substantially degrades speaker similarity (S-SIM drops to 0.807 on CREMA-D and 0.844 on IEMOCAP at $\alpha$=2.0), whereas SLM steering maintains high S-SIM even at $\alpha$=5.0 (0.870 and 0.890).

### F.2. Joint SLM and Flow-Matching Steering

To investigate whether simultaneous steering in both the SLM and flow-matching modules yields additional benefits, we apply steering vectors to both modules jointly on CosyVoice2, using our SLM steering method combined with the EmoSteer-TTS (Xie et al., 2025) approach for the flow-matching module. The same $\alpha$ is applied to both modules, swept from 0.5 to 2.5. Results are compared against SLM-only steering (CoCoEmo) from Table 2.

*Table 9.* Joint SLM + flow-matching steering vs. SLM-only steering (CoCoEmo) on CosyVoice2 for mixed-emotion synthesis.

| Dataset | Method | E-SIM↑ | TEP↑ | $\rho$↑ | H-Rate↑ | S-SIM↑ | WER↓ |
|---|---|---|---|---|---|---|---|
| CREMA-D | Joint $\alpha$=0.5 | 0.751 | 0.063 | 0.050 | 0.679 | **0.865** | 1.12 |
| | Joint $\alpha$=1.0 | 0.767 | 0.131 | 0.112 | 0.695 | 0.859 | **1.02** |
| | Joint $\alpha$=1.5 | 0.784 | 0.198 | 0.178 | 0.715 | 0.832 | 1.32 |
| | Joint $\alpha$=2.0 | 0.787 | 0.163 | 0.176 | 0.711 | 0.808 | 1.06 |
| | Joint $\alpha$=2.5 | 0.781 | 0.152 | 0.143 | 0.700 | 0.770 | 6.54 |
| | CoCoEmo $\alpha$=3.0 | 0.762 | 0.100 | 0.166 | 0.709 | 0.872 | 1.01 |
| | CoCoEmo $\alpha$=5.0 | 0.779 | 0.149 | **0.209** | **0.724** | 0.870 | 0.78 |
| IEMOCAP | Joint $\alpha$=0.5 | 0.906 | 0.197 | 0.101 | 0.718 | **0.889** | **5.71** |
| | Joint $\alpha$=1.0 | 0.912 | 0.237 | 0.193 | 0.746 | 0.884 | 6.05 |
| | Joint $\alpha$=1.5 | **0.915** | 0.266 | 0.246 | 0.763 | 0.869 | 6.93 |
| | Joint $\alpha$=2.0 | 0.911 | 0.274 | 0.170 | 0.737 | 0.845 | 6.29 |
| | Joint $\alpha$=2.5 | 0.907 | 0.300 | 0.081 | 0.709 | 0.801 | 20.93 |
| | CoCoEmo $\alpha$=3.0 | 0.911 | 0.228 | 0.186 | 0.744 | 0.891 | 5.86 |
| | CoCoEmo $\alpha$=5.0 | 0.915 | 0.253 | **0.215** | **0.755** | 0.890 | 6.27 |

As shown in Table 9, joint steering does not improve proportional mixed-emotion control over SLM-only steering. On CREMA-D, the best joint configuration ($\alpha$=1.5) achieves lower $\rho$ (0.178 vs. 0.209) and H-Rate (0.715 vs. 0.724) than SLM $\alpha$=5.0, with substantially worse S-SIM (0.832 vs. 0.870). On IEMOCAP, joint $\alpha$=1.5 reaches a higher $\rho$ (0.246) than SLM $\alpha$=5.0 (0.215), but at the cost of degraded S-SIM (0.869 vs. 0.890) and higher WER (6.93 vs. 6.27). Increasing joint $\alpha$ beyond 2.0 causes severe degradation, with WER reaching 20.93 and S-SIM dropping to 0.770–0.801. These results indicate that additional CFM steering introduces interference rather than complementary information, confirming the advantage of targeted SLM-only steering.

## G. Text-Emotion Mismatch Synthesis Additional Results

We report text-emotion mismatch evaluation results on the IEMOCAP dataset using both CosyVoice2 and IndexTTS2, as summarized in Figure 13 and Table 10. Under low- and mid-mismatch conditions, both models achieve strong E-SIM and TEP scores even without activation steering, indicating that instruction-based control (CosyVoice2) or native emotion

*Table 10.* Evaluation on Low-mismatch and Mid-mismatch sets. Best results are **bold**, second-best are underlined.

| Model | Method | Low-mismatch | | | | Mid-mismatch | | | |
|---|---|---|---|---|---|---|---|---|---|
| | | E-SIM↑ | TEP↑ | S-SIM↑ | WER↓ | E-SIM↑ | TEP↑ | S-SIM↑ | WER↓ |
| CosyVoice2 | No-steer | 0.883 | 0.394 | 0.878 | 7.34 | 0.882 | 0.328 | 0.887 | 7.06 |
| | Instruction | 0.897 | **0.649** | 0.872 | **2.47** | 0.896 | 0.566 | 0.877 | **2.26** |
| | CoCoEmo ($\alpha$=3.0) | 0.901 | 0.515 | **0.882** | 6.08 | 0.901 | 0.446 | **0.888** | 6.80 |
| | CoCoEmo ($\alpha$=6.0) | **0.908** | 0.638 | 0.879 | 7.00 | **0.913** | **0.576** | 0.884 | 6.61 |
| IndexTTS2 | No-steer | 0.876 | 0.377 | **0.875** | 5.80 | 0.875 | 0.326 | 0.881 | **5.31** |
| | Emo_vector | 0.881 | **0.671** | 0.846 | **5.60** | 0.880 | **0.632** | 0.855 | 5.39 |
| | CoCoEmo ($\alpha$=3.0) | 0.889 | 0.514 | 0.872 | 6.46 | **0.888** | 0.456 | **0.882** | 5.42 |
| | CoCoEmo ($\alpha$=6.0) | **0.892** | 0.628 | 0.870 | 6.85 | 0.887 | 0.626 | 0.879 | 7.95 |

conditioning (IndexTTS2) is generally sufficient when textual and emotional cues are largely aligned. As the degree of text-emotion mismatch increases, performance under instruction-only or native conditioning degrades, reflecting the growing influence of text-implied emotional bias. In this regime, activation steering consistently improves E-SIM, with larger steering strengths $\alpha$ yielding greater gains, demonstrating its effectiveness in mitigating semantic–emotional conflicts and biasing acoustic realization toward the target emotion.

IndexTTS2, which is explicitly designed and trained for continuous emotion control, exhibits stronger baseline robustness under high-mismatch conditions compared to CosyVoice2. Nevertheless, activation steering provides complementary improvements in both E-SIM and TEP for IndexTTS2, while preserving speaker similarity (S-SIM) and maintaining comparable WER. These results indicate that activation steering remains beneficial even for models with built-in emotion control mechanisms, particularly in challenging high-mismatch scenarios.

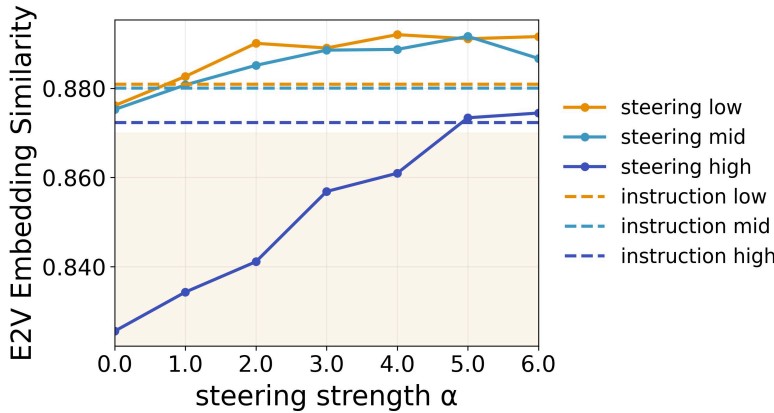

*Figure 13.* E-SIM of high text–emotion mismatch synthesis under low, mid, and high conditions using IndexTTS2 on IEMOCAP. Dash lines: instruction-based control baselines.

## H. Single-Emotion Steering Additional Results

We report single-emotion synthesis results on the in-distribution datasets ESD, RAVDESS, and CREMA-D, as well as the out-of-distribution IEMOCAP dataset, using both CosyVoice2 and IndexTTS2. As shown in Figure 6, increasing the steering strength $\alpha$ consistently improves TEP across target emotions compared to no-steering and instruction-based baselines.

Table 11 summarizes quantitative results across datasets and models. Single-emotion steering improves TEP for both backbones while preserving S-SIM and maintaining comparable WER, including under out-of-distribution evaluation on IEMOCAP. These results confirm that single-emotion steering provides a stable and effective foundation for subsequent mixed-emotion steering experiments.

## I. Single-Emotion Top-K Layer Selection

To select effective steering layers, we perform a layer selection analysis based on the layer-wise steering results. For each backbone, we rank candidate layers by their per-layer steering effectiveness and evaluate steering configurations using the top-$K$ layers ($K = 1$ to $5$). For each configuration, we sweep the steering strength $\alpha$ from 0.5 to 7.0 in increments of 0.5.

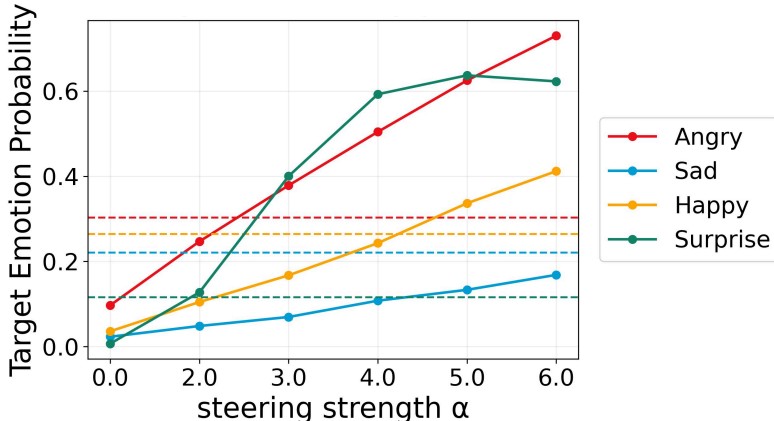

*Figure 14.* TEP for single-emotion synthesis on IndexTTS2 using ESD, RAVDESS, and CREMA-D. Dashed lines: instruction baselines.

*Table 11.* Single-emotion steering evaluation on both in-distribution and out-of-distribution datasets using CosyVoice2 and IndexTTS2. Best results are **bold**, and second-best results are underlined.

| Model | Method | In-Distribution (ESD, RAVDESS, CREMA-D) | | | | Out-Of-Distribution (IEMOCAP) | | | |
|---|---|---|---|---|---|---|---|---|---|
| | | E-SIM↑ | TEP↑ | S-SIM↑ | WER↓ | E-SIM↑ | TEP↑ | S-SIM↑ | WER↓ |
| CosyVoice2 | No-steer | 0.444 | 0.023 | 0.899 | **4.31** | 0.873 | 0.334 | 0.886 | 7.38 |
| | Instruction | **0.600** | 0.278 | 0.886 | 4.50 | 0.890 | 0.575 | 0.877 | **2.51** |
| | CoCoEmo ($\alpha$=3.0) | 0.515 | 0.134 | 0.904 | 4.83 | 0.894 | 0.455 | **0.887** | 6.65 |
| | CoCoEmo ($\alpha$=6.0) | 0.589 | **0.366** | **0.906** | 5.33 | **0.907** | **0.590** | 0.883 | 6.93 |
| IndexTTS2 | No-steer | 0.469 | 0.041 | 0.918 | **4.87** | 0.868 | 0.328 | **0.879** | **5.30** |
| | Emo_vector | 0.518 | 0.226 | 0.718 | 5.70 | 0.880 | **0.639** | 0.853 | 5.36 |
| | CoCoEmo ($\alpha$=3.0) | 0.604 | 0.254 | 0.920 | 4.94 | 0.885 | 0.465 | **0.879** | 5.50 |
| | CoCoEmo ($\alpha$=6.0) | **0.698** | **0.433** | **0.921** | 5.24 | **0.887** | 0.614 | 0.877 | 7.16 |

Steering layers are selected based on a trade-off between emotion controllability and speech intelligibility. Specifically, we impose a WER constraint, requiring that the WER of a steered configuration does not exceed the no-steering baseline by more than +0.5. Among configurations satisfying this constraint, we favor those achieving higher emotion controllability (E-SIM and TEP) and a larger maximum usable steering strength $\alpha_{\max}$, which reflects greater robustness and flexibility of the selected steering sites.

Table 9 summarizes the layer selection analysis. Based on this criterion, we select layers 17 and 14 for CosyVoice2, and layers 6, 8, and 1 for IndexTTS2, which provide a favorable balance between strong emotion control and stable speech quality.

*Table 12.* Layer selection analysis for steering site selection. The no-steering WER serves as the baseline, and only configurations whose WER does not exceed the baseline by more than +0.5 are considered.

| Backbone | Criterion | Top-$K$ Layers | $\alpha_{\max}$ | E-SIM↑ | TEP↑ | S-SIM↑ | WER↓ |
|---|---|---|---|---|---|---|---|
| CosyVoice2 | No-steer (WER = 4.31) | 17 | 5.5 | 0.563 | 0.219 | 0.905 | 4.72 |
| | | 17, 14 | 4.5 | 0.577 | 0.251 | 0.908 | 4.70 |
| | WER-constrained ($\leq 4.31 + 0.5$) | 17, 14, 18 | 4.0 | 0.576 | 0.249 | 0.906 | 4.71 |
| | | 17, 14, 18, 10 | 3.0 | 0.541 | 0.180 | 0.904 | 4.69 |
| | | 17, 14, 18, 10, 11 | 3.5 | 0.568 | 0.223 | 0.905 | 5.76 |
| IndexTTS2 | No-steer (WER = 4.87) | 6 | 7.0 | 0.523 | 0.129 | 0.920 | 5.00 |
| | | 6, 8 | 6.5 | 0.603 | 0.251 | 0.919 | 5.00 |
| | WER-constrained ($\leq 4.87 + 0.5$) | 6, 8, 1 | 6.0 | 0.721 | 0.483 | 0.920 | 5.12 |
| | | 6, 8, 1, 7 | 4.5 | 0.739 | 0.510 | 0.920 | 5.28 |
| | | 6, 8, 1, 7, 2 | 4.0 | 0.746 | 0.527 | 0.921 | 5.15 |

