# OpenReview forum: "CoCoEmo: Composable and Controllable Human-Like Emotional TTS via Activation Steering"
_ICML.cc/2026/Conference — ICML 2026 regular_

### Official Review · Reviewer_tJQA · 2026-02-27

**Soundness:** 3
**Presentation:** 3
**Significance:** 3
**Originality:** 3
**Overall Recommendation:** 5
**Confidence:** 5

**Summary:**

This paper introduces CoCoEmo, a framework for emotional Text-to-Speech (TTS) that uses activation steering to achieve nuanced and composable emotional control without retraining. The researchers demonstrate that emotional prosody is primarily driven by the Speech Language Model (SLM) module rather than the flow-matching decoder, identifying specific mid-to-late layers as the most effective sites for intervention. By injecting weighted direction vectors into these latent layers, the system can synthesize complex mixed emotions and handle text-emotion mismatches more effectively than traditional prompt-based methods. Ultimately, this lightweight approach allows for fine-grained, quantitative control over emotional intensity and blending while maintaining high naturalness and speaker consistency.

**Compliance With Llm Reviewing Policy:**

Affirmed.

**Key Questions For Authors:**

1. Why do different models require different optimal parameters (e.g., the steering strength $\alpha$ and the number of layers $K$), as shown in Table 2? How should a user effectively select these parameters? Specifically, what underlying model characteristics are these parameters correlated with (e.g., hidden dimension or architectural depth), and should they be adjusted dynamically/flexibly rather than remaining fixed across different synthesis tasks?
2. Regarding the layer selection in Table 9, was this determined purely through empirical experimentation? Could you clarify the specific logic behind selecting layers 17 and 14 for CosyVoice2? Since these are clearly not random choices, what specific metrics or representational properties dictated this selection
3. In Table 2, the trends across different metrics (e.g., E-SIM, TEP, and $\rho$) do not always appear consistent. What explains this divergence?
4. Would it be feasible or beneficial to apply activation steering to both the SLM and the Flow-matching module simultaneously?

**Limitations:**

yes

**Strengths And Weaknesses:**

### Strength
1. The paper is well-organized and provides an extensive experimental evaluation. It offers a deep dive into the various factors influencing activation steering, ensuring a thorough understanding of the underlying mechanics.
2. The proposed analysis is demonstrated across multiple state-of-the-art architectures, proving its versatility capability.


### Weakness
1. Figure 1 is dense, making it difficult to interpret at first glance. Specifically, the visualization of the mixed-emotion steering vector is unclear and requires reading the entire manuscript to be fully understood. Simplifying the diagram would significantly improve the paper's accessibility.
2. The "Disentangling" analysis in Section 2 feels inappropriate. Given that prior work (e.g., [1]) suggests these modules are only "softly disentangled," the claim of distinct separation between the SLM and flow-matching modules warrants further justification.
3. The paper argues that steering should primarily occur in the SLM. However, it remains unclear why steering is restricted to a single module; the authors should discuss whether simultaneous steering in both the SLM and the flow-matching module was explored and if it could yield superior results.
4. The methodology in this paper appears similar to that of [2]. A more detailed comparison is needed to clarify the specific technical novelties of this work, particularly regarding the compositional logic used for mixed-emotion steering and the specific identification of optimal steering sites within hybrid architectures.

[1] Zhang, Leying, et al. "DeepASMR: LLM-Based Zero-Shot ASMR Speech Generation for Anyone of Any Voice." arXiv preprint arXiv:2601.15596 (2026).

[2] Xie, Tianxin, et al. "EmoSteer-TTS: Fine-Grained and Training-Free Emotion-Controllable Text-to-Speech via Activation Steering." arXiv preprint arXiv:2508.03543 (2025).

---

> ### Author Rebuttal · Authors · 2026-03-31
>
> We thank Reviewer for the insightful review, and for recognizing our work as “well-organized,” featuring “extensive experimental evaluation,” and providing “a deep dive”.
>
> - **W1: Figure 1 dense.**
>
>     **Response:**  We have simplified Figure 1 by streamlining the steering vector construction section. New figure at: https://anonymous.4open.science/r/cptbtptp/main.png
>
> - **W2: The "disentangling" inappropriate given "soft disentanglement."**
>
>     **Response:**  Thanks for the reference. Section 2 does not claim strict disentanglement. Cross‑conditioning (Table 1) shows the flow‑matching module still carries emotional cues (CCC ≠ 1.0), while the SLM exerts stronger influence. We aim to highlight relative specialization, not strict separation, and will revise Section 2 to replace “disentangling” with “analyzing.”
>
> - **W3 & Q4: Steer SLM and flow-matching simultaneously.**
>
>     **response:** We appreciate this insightful question. Our focus on SLM-based steering is guided by Section 2, which shows that emotion representations are most linearly separable and controllable in the SLM. To address the question, we conducted experiments steering both the SLM and flow-matching simultaneously on CosyVoice2. Simultaneous steering underperforms CoCoEmo while producing similar or worse S-SIM and WER. This is expected as additional steering in the flow-matching module introduce interference, particularly because flow-matching is a suboptimal steering site with lower separability and entangled speaker information (See W4 response). Increasing steering strength across both modules can increase synthesized emotion but does not improve proportional control (ρ/H-Rate) and further degrades S-SIM and WER. Details at https://anonymous.4open.science/r/cptbtptp/SLM_FM_Combined.md.
>
>     | Dataset | Model | E-SIM↑ | TEP↑ | ρ↑| H-Rate↑ | S-SIM↑ | WER↓|
>     | --- | --- | --- | --- | --- | --- | --- | --- |
>     | CREMA-D | combine | 0.767 | 0.131 | 0.112 | 0.695 | 0.859 | 1.02 |
>     |  | CoCoEm | **0.779** | **0.149** | **0.209** | **0.724** | **0.870** | **0.78** |
>     | IEMOCAP | combine | 0.912 | 0.237 | 0.193 | 0.746 | 0.884 | **6.05** |
>     |  | CoCoEm | **0.915** | **0.253** | **0.215** | **0.755** | **0.890** | 6.27 |
>
> - **W4: Comparison with EmoSteer-TTS.**
>
>     **Response**: The technical novelties lie in three aspects: (i) Where to steer (analysis-driven vs. heuristic). We perform systematic analysis (Section 2) to identify the SLM as optimal. EmoSteer-TTS steers flow-matching without justification. Linear probing on flow-matching yields test accuracy 0.61 (train 0.93, chance 0.20) with speaker-disjoint splits, showing emotion and speaker are entangled, whereas SLM achieves ~0.80. Steering in flow-matching is therefore more sensitive to speaker variation and risks degrading quality.
>
>     (ii) How to steer (compositional vs. heuristic). We construct mixed-emotion vectors aligned with ground-truth multi-rater distributions (Section 3), enabling principled blending and handling text–emotion mismatch. EmoSteer-TTS uses fixed heuristic weights without perceptual grounding.
>
>     (iii) How to evaluate. We introduce a multi-rater evaluation framework (ρ, H-Rate, E-SIM) against ground-truth perceptual distributions. EmoSteer-TTS evaluates multi-emotion only via emotion2vec probabilities without ground-truth reference.
>
> - **Q1: Why different K and α.**
>
>     **Response:** Optimal parameters vary due to (a) architectural differences, (b) emotion-conditioning mechanisms (instruction vs. vectors), and (c) pretraining variations. These affect where and how strongly emotion is encoded, making parameters model-dependent (Section 4.5). We provide a systematic selection pipeline (Section 2.2, Appendix I): linear probing ranks layer discriminability → select top-K layers → sweep α under a WER constraint.
>
> - **Q2: Layer selection logic for CosyVoice2**
>
>     **Response:** The selection is principled rather than arbitrary. (i) Layer‑wise emotion discriminability identifies the top-K layers with the highest attn_output emotion separability (section 2); and (ii) a grid search over top‑K layer combinations (K = 1–5), each evaluated with α‑sweeps under a WER‑stability constraint (Table 9 and Appendix I). The resulting top‑2 configuration, layers 17 and 14, achieves the best balance between controllability and speech quality.
>
> - **Q3: Metric divergence in Table 2.**
>
>     **Response:** The divergence reflects a natural trade-off: stronger steering boosts emotion-related metrics (E-SIM, TEP) while slightly affecting speaker/content stability (S-SIM, WER). Each emotion metric captures a different aspect. A method can achieve high TEP yet low ρ if it favors dominant emotions, failing to preserve proportional ranking. CoCoEmo improves ρ, H-Rate, and other metrics by distributing energy proportionally. We will clarify this in the revision.

---

> > ### Author Rebuttal · Reviewer_tJQA · 2026-04-01
> >
> > Thank you for your detailed response. It has addressed most of my concerns and I will raise my score.

---

### Official Review · Reviewer_ZjP8 · 2026-03-12

**Soundness:** 3
**Presentation:** 4
**Significance:** 3
**Originality:** 3
**Overall Recommendation:** 4
**Confidence:** 4

**Summary:**

This paper provides a systematic and comprehensive analysis of the feasibility of activation steering for emotional control in TTS, and proposes a training-free, inference-time lightweight steering approach. In addition, it introduces a series of evaluation metrics to assess the quality of the steering.

**Compliance With Llm Reviewing Policy:**

Affirmed.

**Key Questions For Authors:**

(1) The proposed approach relies on a linear steering assumption, which is helpful for controllability and enables compositional combination. However, it would be interesting to further explore whether emotional information is indeed encoded in such a linear and separable manner within the LLM representations, or whether more complex structures may exist.

(2) Some emotions are inherently correlated with each other. While the paper attempts to disentangle emotion from speaker characteristics and textual semantic information, it remains unclear to what extent emotions can be fully disentangled in practice. In addition, it would be interesting to explore whether the proposed method can generalize to out-of-distribution (OOD) emotions that are not explicitly represented in the predefined emotion set.

(3) It would be helpful to include comparisons with other existing emotion control methods for TTS, both in terms of quantitative metrics and qualitative examples, to better position the effectiveness of the proposed approach.

**Limitations:**

The proposed method relies on high-quality annotated data. In particular, controlling factors such as speaker identity and textual content requires consistent and well-curated annotations, which may limit the applicability of the approach when such high-quality labeled datasets are not available.

**Strengths And Weaknesses:**

Strengths:

(1) The paper presents a relatively comprehensive analysis of where emotional information is encoded in the speech generation process. It further provides a fine-grained investigation at the level of specific layers and operator types.

(2) Introduced several quantifiable evaluation metrics for assessing emotional quality in TTS, enabling a more thorough analysis of generated speech from multiple perspectives.

(3) Steering results are largely consistent with the conclusions derived from the analysis, providing supporting evidence for the authors' claim that activation steering can effectively influence the emotional characteristics of generated speech.

(4) The framework supports compositional emotion control and text--emotion mismatch editing, which expands the potential application scenarios of TTS systems and increases the diversity of controllable outputs.

Weaknesses:

(1) The general analysis and control pipeline based on activation steering has been widely explored in the text and vision domains. As a result, the methodological novelty of applying this paradigm to TTS may appear somewhat incremental.

(2) Emotional control in TTS has been widely studied, the paper provides limited discussion or comparison with other existing emotion control approaches. Most comparisons are conducted only with the base model or relatively simple control strategies.

(3) In Table2, the isolated contribution of activation steering itself is not always clearly observable. In some cases, the improvements may be partly attributable to the instruction or the Emo-vector rather than the steering mechanism alone. However, the proposed method appears to be compatible with other strategies, which suggests it could still serve as a useful plug-in module.

---

> ### Author Rebuttal · Authors · 2026-03-31
>
> We thank Reviewer ZjP8 for recognizing our work as a “comprehensive analysis,” highlighting its “multiple perspectives,” and noting that it “expands the potential application”.
>
> - **W1: Novelty may appear somewhat incremental.**
>
>  **Response:** The novelty lie not only in the extraction, but in three key perspectives: a new analytic task, a compositional steering approach, and a new evaluation framework.
>
>  First, we provide the first systematic analysis (Section 2), revealing where emotion is encoded and where steering should be applied. Second, we construct compositional emotions via weighted steering, with weights tied to ground-truth multi-rater perceptual distributions. This enables principled blending at arbitrary ratios (Section 3) for more human-like emotion synthesis, capabilities absent from prior work. Design in the acoustic latent space independent of text further allows steering to generalize. Third, no prior steering work addresses mixed-emotion or text–emotion mismatch evaluation. We introduce the first multi-rater evaluation framework combining novel metrics (e.g., E-SIM, H-Rate, and ρ), filling a gap in both steering and expressive TTS research. This will be emphasized in the introduction.
>
> - **W2 & Q3: Limited comparison with other emotion control methods.**
>
>  **Response:**  We acknowledge that broader comparisons would strengthen positioning. However, most prior emotion control methods, including label-based (EmoSphere++, EmoDubber) and description-based (EmoVoice, FleSpeech), are designed for acoustic-model/vocoder pipelines and require retraining. Adapting them to hybrid architectures is non-trivial and beyond the scope of this work.
>
> EmoSteer-TTS (Xie et al., 2025) is the only comparable training-free method. Our reproduction on CosyVoice2 shows that CoCoEmo shows comparable and stronger mixed-emotion control than EmoSteer-TTS (ρ: 0.209 vs. 0.072; H-Rate: 0.724 vs. 0.676) while significantly preserving S-SIM and WER. This aligns with our linear-probing analysis: flow-matching representations have lower discriminability than the SLM, with emotion entangled with speaker identity (see Reviewer tJQA W4).
>
> - **W3:  Isolated contribution in Table 2; plug-in nature.**
>
> **Response:** We appreciate the reviewer's observation of CoCoEmo's plug-in nature. The isolated effect of steering is directly observable in the "No-steer → CoCoEmo (α=3.0/5.0)" rows in Table 2, which apply steering without any instruction or Emo-Vector. These rows show steering alone enables mixed-emotion control from scratch. The combined configurations are included to demonstrate the plug-in nature, and their consistent additive gains suggest steering provides complementary information beyond native controls.
>
> - **Q1: Whether emotion is encoded linearly.**
>
> **Response:** We appreciate this suggestion and agree that future work could probe the geometry of emotion representations (e.g., via sparse autoencoders or manifold analysis). As the first systematic study of emotion steering in hybrid TTS, we provide indirect evidence of approximately linear encoding: (i) linear separability analysis (Section 2) shows strong emotion-class separation (~0.8 accuracy); (ii) consistent improvements across emotions, speakers, and datasets indicate that mean-difference steering generalizes, whereas highly non-linear representations would not; (iii) random-noise steering fails (Figure 4), confirming direction-specific effects; (iv) interpolation weights correlate monotonically with Spearman’s ρ (Figure 4c), showing smooth, graded transitions consistent with linear structure. We will clarify this in the revision.
>
> - **Q2-1: Whether emotions can be fully disentangled.**
>
> **Response:** We agree that some emotions are correlated and not perfectly separable. Our goal is practical disentanglement sufficient for controllable manipulation, rather than strict theoretical independence. As shown in Section 4.2, steering along our learned directions consistently shifts emotional expression while preserving speaker identity and intelligibility, indicating that residual entanglement does not impede effective control in practice.
>
> - **Q2-2: Generalizes to OOD emotions.**
>
> **Response**: For OOD emotions, our compositional framework synthesizes nuanced mixed states via weighted combinations of emotion directions, extending beyond the predefined set. While not designed for entirely novel categories, the method is inherently extensible: continuous affect-space steering or unsupervised direction discovery could accommodate OOD emotions. We will incldue this in the discussion.
>
> - **L1: Dependence on high-quality annotated data.**
>
> **Response:** High‑quality data enables more reliable extraction of steering vectors and more stable disentanglement of factors. This is only required during the extraction stage, and can be directly applied to new OOD datasets or domains (Tables 2, 3, 8) without any additional labeling. We will clarify this in the discussion.

---

> > ### Author Rebuttal · Reviewer_ZjP8 · 2026-04-01
> >
> > Thank the authors for their serious response,I still maintain my previous evaluation opinion.

---

### Official Review · Reviewer_TFoh · 2026-03-13

**Soundness:** 3
**Presentation:** 2
**Significance:** 2
**Originality:** 2
**Overall Recommendation:** 4
**Confidence:** 4

**Summary:**

This paper introduces CoCoEmo, a method for controlling emotional expression in modular Text-to-Speech (TTS) systems using steering vectors. The authors conduct a diagnostic analysis to identify that the Speech Language Model (SLM) is the optimal site for steering, propose a method for compositional and mismatch-aware emotion control, and introduce a multi-rater evaluation framework for mixed emotions. While the problem is important and the approach is interesting, the paper suffers from several critical flaws in its experimental design, evaluation methodology, and novelty that undermine its contributions and conclusions.

**Compliance With Llm Reviewing Policy:**

Affirmed.

**Key Questions For Authors:**

The "Random-noise steering" and "Dominant-emotion steering" baselines mentioned in Section 4.1 are diagnostic at best and not reported in the main results (Tables 2, 3). Their absence suggests they were easily outperformed. The main comparison is against the model's "No-steer" output and the native control methods (Instruction for CosyVoice2, Emo-Vector for IndexTTS2). This is a low bar. The key question is not whether steering is better than nothing, but whether it is better than or complementary to other lightweight, post-hoc control methods.

**Limitations:**

1. For IndexTTS2's "Emo-Vector" baseline in mixed-emotion tasks, the authors scale the ground-truth distribution by 0.6, stating that "stronger conditioning was empirically observed to degrade speech intelligibility" (Appendix E.1). However, for CoCoEmo, they are free to tune α to 5.0 or 6.0 to maximize emotion similarity, even if it degrades WER (which it does, see Table 2). The comparison is not held at an equal level of quality. The Emo-Vector baseline should have been tuned to its own optimal trade-off point, not hobbled with a fixed, low intensity.
2.For CosyVoice2's "Instruction" baselines, the paper uses two variants. Instruction2 is an "oracle-assisted" baseline that uses the ground-truth multi-rater percentages in the prompt. This is an unrealistic baseline, as a real-world user would not know these percentages. It serves mainly to be outperformed by CoCoEmo, which also uses this oracle information (Equation 7) to create its steering vector. The comparison is therefore circular.

**Strengths And Weaknesses:**

1. The analysis relies on: 1) a text emotion classifier to get the "text VA," 2) the IEMOCAP annotations for the "audio VA," and 3) an ℓ2 distance between them to define "mismatch." Errors or biases in the text classifier directly contaminate the definition of the mismatch levels. The paper does not provide an analysis of the text classifier's performance on IEMOCAP text, making it impossible to assess the reliability of the high/mid/low mismatch splits. The results could simply reflect that the text classifier is more confident on some sentences than others.
2. The core metric for mixed emotions is Spearman's ρ, which measures the rank-order agreement between the predicted emotion distribution and the ground-truth multi-rater distribution. A high ρ means the model's output has an embedding that is closer to the "happy" centroid than the "sad" one, and so on. This does not measure whether a listener perceives a simultaneous blend of happy and sad. It could simply mean the model produced a slightly happier version of a sad utterance, which would also shift the rank order.

---

> ### Author Rebuttal · Authors · 2026-03-31
>
> We thank Reviewer TFoh for the detailed review and appreciate the recognition of our work as addressing an “important problem” with an “interesting approach.” We have carefully addressed the comments below.
>
> - **W1:  Text VA classifier for mismatch levels.**
>
> **Response:** Direct VA validation on IEMOCAP text is not feasible, as speech datasets provide VA annotations for audio, not text: a field-wide limitation. Our goal is to construct a *relative proxy* for cross-modal inconsistency, not absolute text emotion labels. Systematic biases would shift all samples similarly, minimally affecting relative ranking. We also observe monotonic trends across mismatch levels (Figure 5, Table 7), indicating meaningful cross-modal differences. To reduce noise, we specifically selected the DeBERTa-based model (Christ et al., 2024) (0.82/0.71 for valence/arousal). It is designed for ordinal VA prediction, which also mitigates the boundary uncertainty between mismatch levels. Evaluations over 2,164 samples further average out individual errors. We will include this discussion in Section 4.3.
>
> - **W2: ρ does not measure perceptual blending.**
>
> **Response:** We appreciate the insights. Beyond Spearman's ρ, our objective metrics such as E-SIM and TEP collectively assess the intended mixed-emotion profile. If ρ improvements were mere “slightly shifting rank order”, E-SIM and TEP would remain low. All four metrics (ρ, H-Rate, E-SIM, TEP) improving in concert (Table 2) makes the “false blending” scenario unlikely. While objective–perceptual links remain open in TTS, we present the first evaluation framework for mixed-emotion synthesis.
>
> - **Q1. Missing post-hoc baselines.**
>
> **Response:** CosyVoice2 and IndexTTS2 represent the strongest baselines within the hybrid TTS architecture. Most prior control methods (e.g., EmoSphere++, HED-TTS, EmoVoice) target acoustic-model/vocoder pipelines and require retraining; adapting them to hybrid TTS is non-trivial and beyond this work’s scope.
>
> As the first systematic study of activation steering for emotional control in hybrid TTS, the only comparable training-free method is EmoSteer-TTS (Xie et al., 2025), which operates on flow-matching module. We reproduced it using the same data as CoCoEmo (Table below). CoCoEmo shows comparable and stronger mixed-emotion control than EmoSteer-TTS (ρ: 0.209 vs. 0.098; H-Rate: 0.724 vs. 0.691) while significantly preserving S-SIM with similar WER. This aligns with our analysis: flow-matching representations have lower discriminability than the SLM, with emotion entangled with speaker identity. Detailed results across α are provided in the https://anonymous.4open.science/r/cptbtptp/emosteer.md.
>
> | Dataset | Model | E-SIM↑ | TEP↑ | ρ↑ | H-Rate↑ | S-SIM↑ | WER↓ |
> | --- | --- | --- | --- | --- | --- | --- | --- |
> | CREMA-D | emosteer | 0.767 | 0.097 | 0.098 | 0.691 | 0.858 | **0.76** |
> |  | CoCoEmo | **0.779** | **0.149** | **0.209** | **0.724** | **0.870** | 0.78 |
> | IEMOCAP | emosteer | 0.910 | 0.218 | 0.138 | 0.729 | 0.885 | **6.08** |
> |  | CoCoEmo | **0.915** | **0.253** | **0.215** | **0.755** | **0.890** | 6.27 |
>
>
> - **L1: Emo-Vector baseline hobbled at 0.6 while CoCoEmo tunes α freely.**
>
>  **Response:** 0.6 was chosen as the maximum Emo-Vector scaling that preserves speech intelligibility and speaker characteristics (S-SIM, Table 2). Beyond 0.6, S-SIM degrades substantially (0.865 → 0.830 on CREMA-D, 0.882 → 0.833 on IEMOCAP), making synthesized emotions unreliable. For a fair comparison at equal level, CoCoEmo maintains high S-SIM even at maximum α = 5–6 (0.863 and 0.882), surpassing Emo-Vector at 0.6. Full results are provided in the link (https://anonymous.4open.science/r/cptbtptp/emovector.md) and will be included in the appendix.
> | Dataset | scale | E-SIM↑ | TEP↑ | ρ↑ | H-Rate↑ | S-SIM↑ | WER↓ |
> | --- | --- | --- | --- | --- | --- | --- | --- |
> | CREMA-D | 0.1 | 0.754 | 0.045 | 0.007 | 0.663 | **0.865** | **5.74** |
> |  | 0.3 | 0.761 | 0.062 | 0.045 | 0.672 | 0.863 | 5.81 |
> |  | 0.6 | **0.767** | 0.165 | 0.236 | 0.731 | 0.850 | **5.74** |
> |  | 0.8 | 0.757 | **0.246** | **0.291** | **0.754** | 0.830 | 5.80 |
> | IEMOCAP | 0.1 | **0.891** | 0.202 | 0.085 | 0.715 | **0.882** | 5.24 |
> |  | 0.3 | 0.889 | 0.245 | 0.184 | 0.746 | 0.877 | 5.06 |
> |  | 0.6 | 0.855 | 0.296 | 0.254 | 0.767 | 0.854 | **4.74** |
> |  | 0.8 | 0.839 | **0.302** | **0.291** | **0.779** | 0.833 | **4.74** |
>
> - **L2: Instruction2 comparison is circular; using oracle percentages.**
>
>  **Response:**  Oracle percentages are a measurement requirement, not a performance advantage. Both systems receive identical information for fair comparison. Without ground-truth percentages, no objective metric can be computed.  CoCoEmo does not require oracle inputs at inference; users can specify any blend ratio. Crucially, Instruction2 (with oracle) underperforms Instruction1 (without oracle) in Table 2, confirming oracle percentages confer no advantage. We will clarify in Section 4.2.

---

> > ### Author Rebuttal · Reviewer_TFoh · 2026-04-06
> >
> > I appreciate your reply, but I still believe the contribution, novelty, and extra experimental data offered do not justify a higher score. As a result, my original rating stands.

---

### Official Review · Reviewer_ZmYV · 2026-03-14

**Soundness:** 3
**Presentation:** 3
**Significance:** 3
**Originality:** 3
**Overall Recommendation:** 3
**Confidence:** 3

**Summary:**

The paper proposes CoCoEmo, a training-free, inference-time activation steering method designed to modulate the emotional expressiveness of hybrid TTS models. Using cross-conditioning ablations and linear probing, the authors demonstrate that emotional prosody is primarily governed by the Speech Language Model rather than the continuous flow-matching decoder, and that mid-to-late attention layers offer the highest linear separability for emotion. The method computes contrastive mean-difference vectors between emotional and neutral speech representations and injects them as a global sequence bias to generate mixed or text-mismatched emotions.

**Compliance With Llm Reviewing Policy:**

Affirmed.

**Key Questions For Authors:**

NA

**Limitations:**

Yes

**Strengths And Weaknesses:**

### Strength
* The cross-conditioning diagnostic effectively isolates the SLM as the primary driver of emotional prosody. This division of labor between the discrete SLM and the continuous flow-matching module is an important insight for the generative speech community.

* The authors systematically use linear probing to map the emotional manifold across layers and operators, identifying attention outputs in mid-to-late layers as optimal steering sites.

### Weakness
On Method:

* The contrastive mean-difference vector extraction fundamentally binds the model to discrete, mutually exclusive categorical labels (e.g., Happy, Sad, Angry) extracted from rigidly paired datasets. It cannot reliably generalize to complex, free-form text prompts or continuous affective states.
* The steering vector is formulated as a static scalar multiplied by a global vector, which is applied uniformly across the entire token sequence. Because emotion is locally controlled in human speech, this global bias fails to offer fine-grained control over dynamic emotional shifts at specific timestamps.

On Evaluation:
* The subjective evaluation relies almost entirely on Naturalness MOS. N-MOS only evaluates acoustic fluency; it cannot decouple naturalness from emotion faithfulness.
* On the OOD IEMOCAP dataset, the method is low-performing. N-Mos of steering (4.06 - 4.33 in IndexTTS2) falls behind the no-steer baseline (4.37 in IndexTTS2)

---

> ### Author Rebuttal · Authors · 2026-03-31
>
> We thank Reviewer ZmYV for the thoughtful review and for recognizing our work as providing “important insights for the generative speech community” and a “systematic” analysis.
>
> - **W1: Mean-difference extraction cannot generalize to continuous affective states or free-form text.**
>
>     **Response:** We clarify that discrete labels are required only at the vector *extraction* stage, not during inference. The extracted vectors reside in a continuous latent space and support continuous control via: (1) smooth intensity interpolation through α (Eq. 8), and (2) compositional combination of multiple vectors at arbitrary weights (Eq. 7), producing a continuous spectrum of blended emotions. Analogous findings in text LLMs suggest that emotional representations tend to be directionally encoded and steerable in latent space (Reichman et al., 2025), motivating the hypothesis that contrastive extraction is geometrically reasonable.
>
>     Regarding free-form text: as confirmed in the out-of-distribution evaluation in Table 2, emotional steering transfers reliably across diverse, naturalistic text inputs not present in the contrastive extraction set. This is due to our method extracting variations primarily in the acoustic latent space of the TTS module (Section 3.1), independent of text content, and therefore generalizing naturally to unseen inputs.
>
>     We acknowledge these insightful suggestions and are happy to include the reference and explicitly acknowledge this boundary condition in a revised limitations section.
>
>     [1] Reichman, B., et al. (2025). Emotions Where Art Thou: Understanding and Characterizing the Emotional Latent Space of Large Language Models. *arXiv*.
>
> - **W2: Static global bias cannot capture dynamic emotional shifts.**
>
>     **Response:** We clarify that while the steering vector is injected uniformly,  its effect is not uniform across tokens due to the model’s autoregressive and attention-based dynamics. In autoregressive generation, each steered token causally influences subsequent tokens through attention, so the steering interacts with sequential generation dynamics rather than acting as a simple additive bias. This is consistent with findings in LLM activation steering (Turner et al., 2024; Zou et al., 2023; Rimsky et al., 2024), where global latent shifts successfully guide autoregressive generation precisely because of this causal propagation. Our utterance-level control aligns with existing TTS approaches (CosyVoice2, IndexTTS2) and the annotation granularity of widely used datasets (CREMA-D, IEMOCAP).  Explicit timestamp-level modulation would introduce additional complexity and potential instability. We consider this a promising future direction when the finer-grained annotated data becomes available. We will clarify this in the revised discussion section.
>
>
> - **W3: N-MOS cannot decouple naturalness from emotion faithfulness.**
>
>     **Response:** We clarify that N-MOS is not our primary evaluation of emotion, it serves to verify that steering preserves speech quality. Emotion faithfulness is comprehensively evaluated through our objective metrics: E-SIM measures overall emotional proximity via Emotion2Vec embeddings, TEP measures recognition confidence across target emotions, ρ and H-Rate assess whether compositional blending weights are faithfully reflected in the output. These four metrics are specifically designed to capture different facets of emotion control effectiveness, which is a core contribution of our work (Section 3.3).
>
> - **W4: On OOD IEMOCAP, steering N-MOS falls behind the no-steer baseline for IndexTTS2.**
>
>     **Response:**  The slight N-MOS reduction especially in the OOD dataset reflects a well-documented expressivity–naturalness trade-off in expressive TTS that any affective perturbation in latent space incurs a small naturalness cost. This is explicitly calibrated via α (Appendix I). Combined with improvements in objective metrics (e.g., emotional alignment and controllability) and strong in-distribution performance across all metrics including N-MOS (Table 2, CREMA-D), the results indicate that our method maintains competitive naturalness while substantially enhancing emotional expressiveness.

---

> > ### Author Rebuttal · Reviewer_ZmYV · 2026-04-04
> >
> > Thanks for the rebuttal. My following concerns are not resolved.
> >
> > 1. unclear if it generalizes to complex, free-form text prompts or continuous affective states
> > 2. unclear if it does fine-grained control over dynamic emotional shifts at specific timestamps.
> > 3. using objective evaluation metric as the primary evaluation method

---

> > > ### Author Response · Authors · 2026-04-05
> > >
> > > We appreciate the reviewer’s time and have provided the detailed response below.
> > >
> > > - **Q1. Generalizes to free-form text prompts or continuous affective states**
> > >
> > > **Response:** Regarding free-form text prompts: we address two interpretations. i) If this refers to free-form text inputs for speech synthesis, our OOD evaluation on IEMOCAP (Section 4.2 and our previous rebuttal) demonstrates reliable generalization. ii) If this refers to free-form text prompts as a control interface, as the instruction-following paradigm in CosyVoice2, our method does not directly address this, as steering operates in the latent space rather than via text conditioning. However, this could be extended by training a lightweight classifier that maps free-form text prompts to soft emotion proportions for steering.
> > >
> > > Regarding continuous affective states, while CoCoEmo does not explicitly target continuous valence-arousal (VA) dimensions as numerical regression targets, our strong performance on IEMOCAP, annotated with both categorical emotion labels and continuous arousal and valence ratings spanning the full [1, 5] range,  suggests the method generalizes across the VA space in practice. Directly handling continuous affective states would require extracting steering vectors from numerical VA annotations rather than categorical labels. A principled extension would replace categorical mean-difference extraction with regression-direction steering, fitting a linear regression probe to predict VA ratings directly from SLM activations and using the weight vector as the steering direction, which [1] formally proves is a valid steering direction for any linearly decodable concept.
> > >
> > > As our focus is on first establishing where emotion is encoded in hybrid TTS architectures and how to achieve training-free compositional steering in the latent space, free-form text prompt control and continuous affective state modeling are beyond the primary scope of this work. We will discuss both as future directions in the revised version. We welcome any further clarification if the question was intended in a different sense.
> > >
> > > [1] Park, Kiho, et al. "The linear representation hypothesis and the geometry of large language models.” ICML 2024.
> > >
> > > - **Q2. Fine-grained control over dynamic emotional shifts at specific timestamps.**
> > >
> > > **Response:** In addition to our previous clarification that utterance-level steering can enable token-wise emotion control due to the autoregressive nature of speech-language models, we further note that dynamic emotional shifts can be explicitly modeled within our framework. For instance, we can apply distinct steering vectors to different segments of a sentence during SLM generation. While this is beyond the scope of the work, we have piloted this extension and provide examples in https://anonymous.4open.science/r/cptbtptp/segement-level/segement_level_steering_results.md. We present three synthesised examples of dynamic emotion shifts (e.g., happy → sad → angry) and show that a downstream Emotion2Vec classifier correctly identifies the intended emotion in each synthesized speech segment with high confidence (scores: 0.997, 0.984, 1.000 for Sample 1), with mel-spectrograms and F0 contours provided as visual reference. If permitted during the rebuttal phase, we would be happy to also share the synthesized audio samples via an anonymous link.
> > >
> > > For more precise token-level control, determining where to apply control and how to prevent steering from affecting subsequent generation requires further consideration in future work. We acknowledge that fine-grained temporal control is an interesting future direction beyond the scope of this work, and will include this discussion.
> > >
> > > We welcome any further clarification should the reviewer have specific scenarios in mind.
> > >
> > > - **Q3. Using objective evaluation metric as the primary evaluation method**
> > >
> > > **Response:** Our evaluation framework combines objective metrics (E-SIM, TEP, H-Rate, and Spearman's ρ) and subjective metrics (N-MOS). Emotion faithfulness is comprehensively captured by our objective metrics, which constitute the first objective evaluation framework for mixed-emotion synthesis, leveraging a multi-rater ground-truth framework to assess whether the intended mixed-emotion profile is acoustically realized beyond dominant emotion accuracy alone. N-MOS serves specifically to verify that steering preserves speech naturalness. Results across both in-distribution (CREMA-D) and OOD (IEMOCAP) datasets consistently demonstrate strong performance, with the slight N-MOS reduction on OOD data reflecting a well-documented expressivity-naturalness trade-off explicitly calibrated via α (Appendix I).
> > >
> > > We acknowledge that broader subjective evaluation including subjective perceived emotion faithfulness would provide more evidence. We will include such evaluations in future studies upon new ethical approval and will note this as future work in the revised version.

---

### Decision · Program_Chairs · 2026-04-30

**Decision:**

Accept (regular)

**Comment:**

This paper presents CoCoEmo, a training‑free, fine‑grained emotion‑controllable text‑to‑speech framework that exploits the activation steering component in the language‑model (LLM) part of modern hybrid TTS systems. The authors first conduct a systematic analysis of hybrid architectures to identify where emotional cues are most linearly separable and most amenable to steering (Section 2). Guided by that analysis, they design a compositional steering strategy that generates mixed‑emotion vectors aligned with empirical multi‑rater emotion distributions (Section 3). Finally, they propose a richer evaluation protocol (including proportional‑ranking metrics such as ρ and H‑Rate) and demonstrate gains over the recent baseline EmoSteer‑TTS across several state‑of‑the‑art models and datasets.

Overall, the work is technically sound: the experimental protocol is thorough, the analysis is reproducible, and the evaluation methodology is well grounded in perceptual ratings. In response to a reviewer’s suggestion, the authors conducted combined steering in both modules and found it to degrade quality, confirming the advantage of their targeted approach.

### Reasons to Accept
1. **Empirical evidence** that steering in the SLM module yields better emotion preservation, proportional ranking, and speaker stability than the recent baseline.
2. **Methodological contributions**: an analysis‑driven module selection, a compositional mixed‑emotion steering strategy, and a richer evaluation protocol.
3. **Comprehensive evaluation** across datasets and architectures, including real‑time ASR‑controlled scenarios.
4. **Novelty** -- while activation steering is not novel, the way the model is integrated into the TTS domain is somewhat new and interesting (despite EmoSteer-TTS)

### Weaknesses
1. **Thoroughness** -- some reviewers (esp ZmYV) ask for additional experiments and potential extensions (e.g. with text-based conditioning, rather then embeddings), which would further enhance the impact of the paper
2. **Protocols** -- some reviewers (ZmYV, TFoh, ZjP8) request more comparisons and more detailed analysis, which the authors provide in a response that seems to cover the requested information

Given the solid experimental foundation, clear (if somewhat limited) methodological novelty (confirming that steering can also be applied to audio), and the additional insights presented during the review process, the AC feels that the paper should be presented, if possible, at ICML. The work makes a worthwhile contribution to the field of controllable TTS and will be useful to practitioners in the field.